⊚ | **Open Peer Review** | Clinical Microbiology | Research Article

# Implementation and validation of a new qPCR assay to detect imported human *Plasmodium* species

Camille Cordier,[1,2] Samia Hamane,[1] Théo Ghelfenstein-Ferreira,[1,3] Sarah Dellière,[1,3] Élodie Da Silva,[1] Blandine Denis,[4] Sandrine Houzé,[5,6] Valentin Joste,[5,6] Alexandre Alanio[1,3]

**ABSTRACT** Diagnosis of imported malaria is based on microscopic examination of blood smears (BS), detection of circulating plasmodial antigen by immunochromatography (ICT), or detection of *Plasmodium* spp. DNA by loop mediated isothermal amplification. We have developed duplex (*Plasmodium* spp. and *Plasmodium falciparum*) real-time PCR (qPCR) and species-specific qPCR assays for the identification and quantification of human *Plasmodium* species. Whole nucleic acids from 523 samples prospectively collected from 410 patients suspected of malaria between June 2016 and March 2021 were tested by qPCRs and compared to standard diagnostic procedures. All qPCRs were designed on 18S ribosomal ribonucleic acid. The limit of detection of duplex qPCR was 8 copies/reaction, and analytical specificity of species-specific qPCRs was 100%. Seventy-nine patients diagnosed for single species malaria involving *P. falciparum*, *P. vivax*, *P. ovale*, and *P. malariae* were positive with duplex and species-specific qPCRs. *P. knowlesi* qPCR detected DNA from a *P. knowlesi* culture. Of eight cases of mixed *Plasmodium* species infection, five were identified with our qPCR assays with better sensitivity as compared to BS and ICT. Eight out of 323 patients with negative BS were hospitalized for symptoms suggestive malaria after malaria-endemic area travel or known with history of malaria had a low positive duplex qPCR. Between day of diagnosis and post-treatment follow-up at D2-D4 of malaria, a 3.1-log *P. falciparum* load decrease was observed by qPCR. These new *Plasmodium* qPCRs allowing detection of human plasmodial species have excellent species specificity and allow rapid detection of *P. falciparum* cases, malaria with low parasitaemia, and mixed *Plasmodium* species infection.

**IMPORTANCE** Malaria is a disease transmitted by a *Plasmodium* parasite genus. Most cases are caused by *Plasmodium falciparum*. Despite a significant drop of incidence and mortality since 2000, 249 million cases and 608,000 deaths have been estimated in 2022, mainly in Africa. Due to the increasing number of travels to endemic areas, incidence of imported malaria is rising in Europe. Various techniques are used in European laboratories, such as microscopic examination of thin and thick smears and rapid diagnostic tests. However, these techniques require skilled operators to differentiate plasmodial species and have limited sensitivity. Actually, molecular diagnosis is carried out using point-of-care test for rapid results with excellent sensitivity but is unabled to determine involved species and assess parasitaemia. In this study, we developed a combined molecular tool based on both detection of all human plasmodial species (*Plasmodium* spp.) and *P. falciparum*. We have also developed specific qPCRs for each human plasmodial species.

**KEYWORDS** malaria, qPCR, duplex, *Plasmodium*, diagnosis

Malaria is a vector-borne disease transmitted by a protozoan parasite of the genus *Plasmodium*. Most human cases are caused by *Plasmodium falciparum* and *Plasmodium vivax*. Other species involved are *Plasmodium ovale curtisi*, *Plasmodium ovale*

**Peer Reviewer** Abhinav Sinha, National Institute of Malaria Research, New Delhi, Delhi, India

Address correspondence to Camille Cordier, camille2.cordier@chu-lille.fr, or Alexandre Alanio, alexandre.alanio@pasteur.fr.

The authors declare no conflict of interest.

*wallikeri*, *Plasmodium malariae,* and *Plasmodium knowlesi*. Approximatively half of the world's population, spread across 85 countries and territories, is at risk of infection (1). Between 2000 and 2015, malaria incidence declined by 37%. Mortality rate decreased by 60% among all age groups and by 65% among children under 5 years (2). Despite this, 249million cases and 608,000 deaths were estimated in 2022, mostly in Africa and among children under 5 years. *P. falciparum* is involved in most severe malaria cases (3). Due to the increasing number of travels to malaria-endemic areas, the number of imported malaria cases in Europe is increasing (4). According to the European Center for Disease Prevention and Control (ECDC), metropolitan France has the highest burden of imported malaria cases in Europe, with an estimated 5,540 cases in 2019 (5). This represented a 6.3% increase as compared to 2017 (Annual Activity Report 2020, French National Reference Center of malaria). The predominant species diagnosed was *P. falciparum* (87.8%).

Different techniques are recommended for the routine diagnosis of malaria in the laboratory. The reference method is still the visualization of ring-stage trophozoïtes by microscopic examination of blood smears (BS) by light microscopy. Microscopy allows both the species diagnosis and estimation of the parasitaemia in order to evaluate the severity of the malarial episode. However, the use of thin smears presented a limited sensitivity of 100 parasites/µL of blood and required a skilled operator (6). Indeed, it is sometimes difficult to detect low parasitaemia and to differentiate some plasmodial species on the basis of their morphological characteristics, such as *P. ovale* and *P. vivax* (7). Correct identification of the species is essential to treat the patient with appropriate drug and to avoid relapses. Contrariwise, the thick smear overcomes the lack of sensitivity of the thin smear, with an estimated sensitivity of 10 parasites/µL of blood. However, this technique is difficult to use for species diagnosis in non-expert centers (8).

Immunochromatographic rapid diagnostic tests (ICT) are used in addition to microscopy and are fast and easy to use, but they are associated with lower sensitivity and specificity especially in non-*P. falciparum* malaria cases or in mixed infections (9).

In many French laboratories, the molecular diagnosis of malaria is performed using the loop-induced isothermal amplification (LAMP) based by Alethia malaria assay (Meridian Bioscience Inc., Cincinnati, OH, USA). LAMP allows for rapid results and has optimal (~100%) sensitivity. However, this LAMP assay is not able to identify the species involved, assess parasitaemia, and identify mixed *Plasmodium* species infection (10). In recent years, many polymerase chain reaction (PCR) tests have been developed to detect, identify, and quantify the different species of *Plasmodium* in blood (11, 12). To date, no real-time quantitative PCR (qPCR) has been developed to detect and discriminate the five plasmodial species involved in human pathology and to quantify the number of plasmodial circulating copies.

A *Plasmodium* duplex qPCR assay based on the detection of a pan-plasmodial and a *P. falciparum* specific locus, in addition to the internal control, has been developed and validated. It was designed to allow the detection of all five plasmodial species and the immediate identification of the presence of *P. falciparum* at the same time. Simultaneously, specific qPCRs to *P. vivax*, *P. ovale* (*curtisi* and *wallikeri*), *P. malariae,* and *P. knowlesi* have been developed. These tests are designed to perform species diagnosis, quantify parasitaemia, and detect low parasitaemia and mixed *Plasmodium* species infection cases on patient blood samples. The qPCR results were compared with those obtained by BS and ICT.

## MATERIALS AND METHODS

### Patients and specimens

Left over whole blood specimens from 410 patients suspected of malaria collected from June 2016 to March 2021 were prospectively processed in parallel to our routine diagnostic procedure (thick and thin blood smear [BS] and BinaxNow point of care immunoassay (Abbott) [ICT]). These patients were admitted to emergency department

of Saint-Louis hospital, Paris, France, for malaria suspicion in febrile patients returning from malaria-endemic areas.

The parasitaemia was calculated for positive thin smear specimens. Malarial episode was defined as the presence of ring-stage trophozoïtes on blood smears with species identification based on the parasite forms morphology on thin smear. The microscopy result was used as the gold standard to study the results obtained with our qPCRs. Microscopic examination was performed by double reading. In case of discrepancy, a medical senior biologist resolved the case.

The ICT can detect the Histidine-Rich Protein II (HRP2) antigen specific of *P. falciparum* (T1) and aldolase (T2), a common antigen of all *Plasmodium* species.

## Species confirmation

These blood samples were referred to the French National Reference Center of malaria for species confirmation by qPCR. The *Plasmodium* Typage (Bio-Evolution, Bussy-Saint-Martin, France) real-time qPCR kit has been used for simultaneous identification of *P. falciparum*, *P. ovale*, *P. vivax*, *P. malariae,* and *P. knowlesi* targeting *18S rRNA* gene with sensitivity of 10 copies/µL. In case of *P. ovale* target positivity, subspecies identification was carried out using in-house qPCR with high-resolution melting (13).

All diagnostic, clinical, and therapeutic data were collected in the French National Reference Center of malaria database (https://cnr-palu.voozanoo.net/palu/).

## Nucleic acids extraction

Whole nucleic acids (WNAs) were extracted from 1.3mL of EDTA whole blood with the addition of 10µL of 1:5 diluted internal control per sample (DNA Virus Culture, DICD-CY-L100, Diagenode, Seraing, Belgium) using a Qiasymphony (Qiagen, Hilden, Germany) with the Virus Pathogen MIDI extraction kit (Qiagen, Hilden, Germany) following manufacturer's instructions. An internal control added before the extraction step was used to control all the qPCR process as recommended by the MIQE guidelines (14). WNAs were eluted in 100µL. All extracts were stored (from 1week to 5 years) at −80°C until use.

## Selection of target gene

The ribosomal small subunit of ribonucleic acid (18S rRNA) gene of *Plasmodium falciparum*, *Plasmodium vivax*, *Plasmodium ovale curtisi*, *Plasmodium ovale wallikeri*, *Plasmodium malariae,* and *Plasmodium knowlesi* together with that of *Babesia divergens*, *Babesia microti,* and *Toxoplasma gondii* were downloaded from the GenBank Database (https://www.ncbi.nlm.nih.gov) and aligned to design accordingly probes and primers using Geneious Prime version 2020.2.4 software (Biomatters Ltd., New Zealand) (Fig. 1; Fig. S1). Probes and primers were listed in Table 1. The absence of hairpins, self-dimers, and hetero-dimers was checked for each primer and probe (https://eu.idtdna.com/). The *in silico* specificity of the primers and probes was tested using primer-BLAST (https://www.ncbi.nlm.nih.gov/tools/primer-blast/).

## Quantitative PCRs

WNAs amplification was performed using the following conditions: 1× of LightCycler 480 Probes Master (Roche Diagnostics GmbH, Mannheim, Germany), 6mM of magnesium sulfate ($MgSO_4$, Invitrogen), 2% of dimethyl sulfoxide (DMSO, Sigma, Life Science), 0.3µM of each primer, 0.1µM of each probe, and 1.75µL of internal control primers and Cy5 probe (DICD-CY-L100, Diagenode, Seraing, Belgium) in a total volume of 35µL including 8µL of eluted nucleic acids. The amplification consisted of one activation step at 95°C for 10 min and 50 cycles of denaturation at 95°C for 15 s and annealing at 58°C for 30 s. All qPCR runs were performed on a Light Cycler 480 thermocycler (LC480-II, Roche Diagnostics, Mannheim, Germany) with Cq determination using the calculation of the second derivative of the amplification curve.

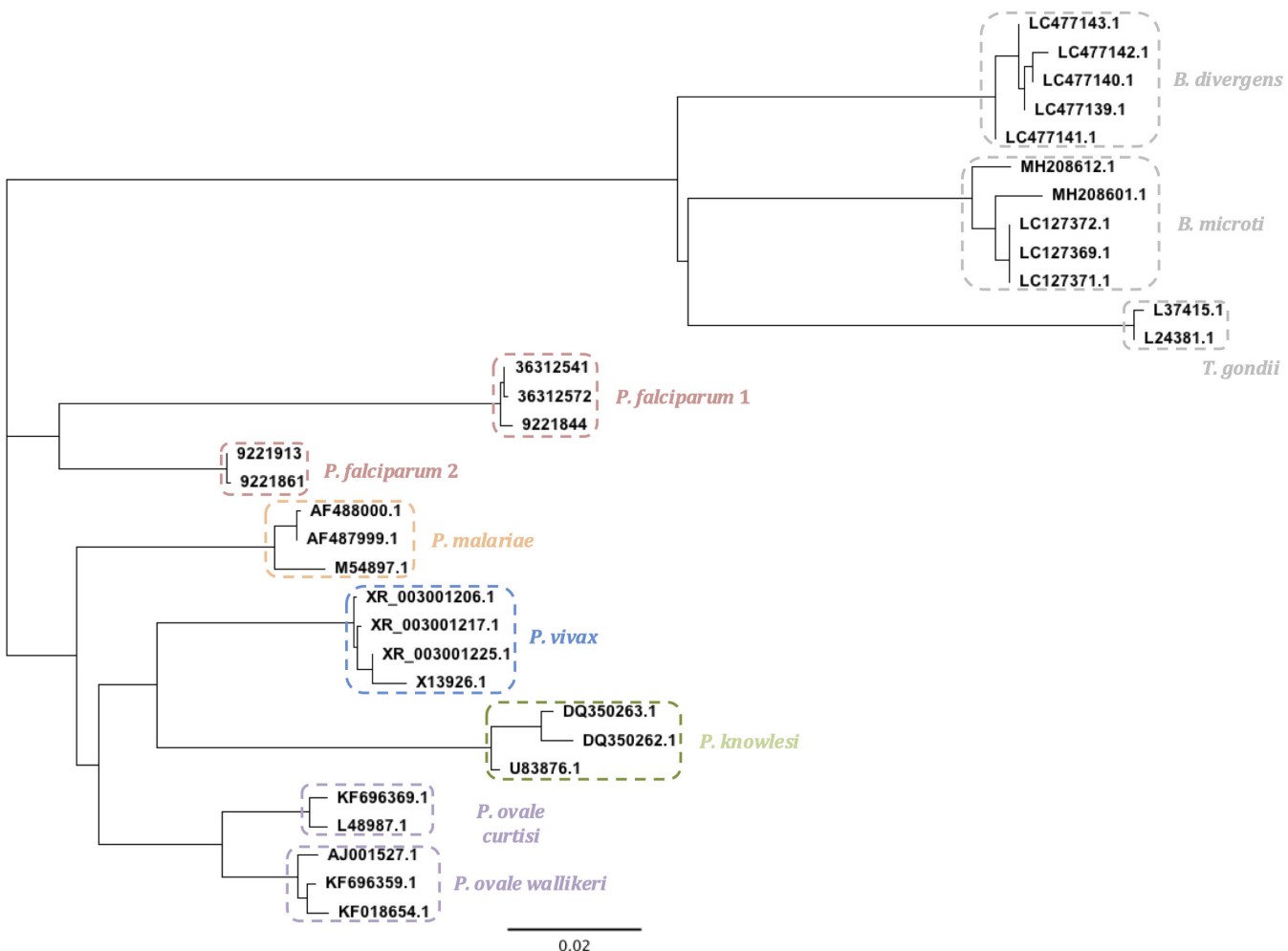

**FIG 1** Phylogenetic tree of the partial 18S rRNA sequences of the five species of malaria parasites infecting humans (*P. falciparum*, *P. vivax*, *P. ovale*, *P. malariae,* and *P. knowlesi*) and three phylogenetically related blood parasites (*B. divergens*, *B. microti,* and *T. gondii*). Several genomic sequences (identified by their gene ID, https://www.ncbi.nlm.nih.gov/gene) for species belonging to the *Apicomplexa* phylum have been aligned in order to study the genetic diversity within each species. For *P. falciparum*, two genetic lineages were observed (*P. falciparum* 1 and 2). The separation of the two *P. ovale* subspecies (*P. ovale curtisi* and *P. ovale wallikeri*) was verified. Phylogenetic tree was built following the Neighbor Joining method correcting with Tamura-Nei method.

Using a conserved *Plasmodium* locus across species of *18S rRNA* gene in a control-plasmid to implement calibration curve, we were able to evaluate the parasite load in copy numbers for pan-*Plasmodium* qPCR. The formula used to define the number of copies from the Cq obtained by qPCR is as follows: $Q = 1{,}000 \times 1.70^{(33.98 - Cq)} = X$ copies/µL ($Q$ = number of copies/µL; 1,000 = number of copies obtained with the control-plasmid at a Cq of 33.98; $E$ = 1.70 = efficiency of qPCR *Plasmodium* spp.; Cq = cycle quantification obtained with qPCR *Plasmodium* spp.).

Parasite load for species-specific qPCRs was expressed in cycles of quantification (Cq) in the absence of species-specific control-plasmid. Parasitaemia obtained by counting blood-stage parasite forms on thin smears was expressed as the number of parasites/µL.

We first validated each single assay on species-specific samples together with our internal control assay and then optimized a duplex assay based on *Plasmodium* spp. (FAM), *P. falciparum* (HEX), and internal control (Cy5) primers and probes.

## qPCR efficiency and limit of detection

The standard curve allowing qPCR efficiency calculation was obtained based on the result of two replicates of eight fivefold serial dilutions of eluted nucleic acids in DNA

**TABLE 1** Sequences of the probes, forward and reverse primers designed on 18S rRNA[a]

| Assays | *Plasmodium* species | 5′ to 3′ sequence |
| --- | --- | --- |
| Duplex qPCR | *Plasmodium* spp. | PanF1 : TTCAGTACCTTATGAGAAATCAAA |
| | | PanR1 : TTAACTTTCTCGCTTGCG |
| | | PanP1 (FAM) : CTTTGGGTTCTGGGGCGA |
| | *P. falciparum* | PanF2 : GCTCCAATAGCRTATATTAAAATT |
| | | Pf1R2 : TATTTGGTTTTCCCAAACC |
| | | Pf2R2 : AGCTAAAATAGTTCCCCTAGAATAG |
| | | PanP2 (HEX) : TTGCAGTTAAAACGYTCGTAGTTGAA-TATT |
| Species-specific qPCRs | *P. vivax* | PanF2 : GCTCCAATAGCRTATATTAAAATT |
| | | PvR2 : TAGGTAGGATGCGCACAG |
| | | PanP2 (HEX) : TTGCAGTTAAAACGYTCGTAGTTGAA-TATT |
| | *P. ovale* | PoF1 : GAAAGATTTTTAAATAAGAAAATTCC |
| | | PanR1 : TTAACTTTCTCGCTTGCG |
| | | PanP1 (FAM) : CTTTGGGTTCTGGGGCGA |
| | *P. malariae* | PmF1 : GATGATAGAGTAAAAAATAAAAGAGAC |
| | | PanR1 : TTAACTTTCTCGCTTGCG |
| | | PanP1 (FAM) : CTTTGGGTTCTGGGGCGA |
| | *P. knowlesi* | PanF2 : GCTCCAATAGCRTATATTAAAATT |
| | | PkR2 : CATAAAGCAGAAAACATATATTGG |
| | | PanP2 (HEX) : TTGCAGTTAAAACGYTCGTAGTTGAA-TATT |

[a]Two reverse primers were developed to amplify the two distinct *P. falciparum* lineages (*P. falciparum* 1 and 2, Fig. 1). One primer pair was designed on a common genomic region allowing amplification of both *P. ovale curtisi* and *P. ovale wallikeri* (with no distinction between them).

(deoxyribonucleic acid) free water for each plasmodial species. The eluates of four positive clinical specimens with parasitaemia between 8,900 and 12,460 parasites/µL for *P. falciparum*, *P. vivax*, *P. ovale*, *P. malariae,* and DNA extract of *P. knowlesi* culture (1.1ng/mL) have been used to study the efficiency and the limit of detection for each qPCR assay.

A 179-base pair DNA plasmid containing only the PCR target locus of the qPCR *Plasmodium* spp. was synthetized at 40ng/µL (Eurogentec, Seraing, Belgium), diluted at different concentrations (1,000 to 1 copies/µL per well), and tested in 4 to 10 replicates to obtain the limit of detection and absolute quantification on LC480 thermocycler (Roche Diagnostics).

## Analytical specificity of qPCR

The development and evaluation of the analytical specificity of different qPCRs were studied on 38 samples positive for a single plasmodial species by conventional methods (BS and ICT), from 38 different patients at the time of diagnosis. Among these samples, 18 were positive for *P. falciparum*, 7 for *P. vivax*, 10 for *P. ovale*, and 3 for *P. malariae*. The *P. knowlesi* DNA was obtained from ongoing *in vitro* cultures of Robert Moon team at LSHTM (London School of Hygiene & Tropical Medicine) (15).

## Validation of qPCR

A total of 190 samples from 87 different patients were used to validate the different qPCRs. These samples were positive for one or two plasmodial species. Of these positive samples, 87 and 103 were collected at the day of diagnosis (D0) and at the time of post-malaria treatment follow-up, respectively. Samples from *P. falciparum*-positive patients were analyzed using the duplex qPCR assay. Samples positive for *P. vivax*, *P. ovale,* and *P. malariae* were analyzed using duplex qPCR and species-specific qPCR assays.

In addition, 333 samples negative by BS from 323 patients suspected of malaria were screened with duplex qPCR assay (Fig. S2).

## Statistical analysis and graphs

Medians and interquartile ranges (IQRs) are given for specific descriptions in non-normally distributed parameters. Contingency tables were performed to analyze the statistical links between clinical presentation, microscopic examination, and qPCR test result. Graphs were obtained using RStudio (version 1.2.5042) with ggplot2 package.

Regression lines were constructed automatically by plotting the logarithm of the initial template concentration vs the corresponding Cq value by using the Analysis package included in LightCycler 480 software version 1.5 (Roche Diagnostics).

## RESULTS

### Limit of detection and efficiency

The limit of detection of the *Plasmodium* spp. assay was 1 copie/µL (Table S1).

Using clinical specimens, the limit of detection was 0.02 parasites/µL for *P. falciparum* and *P. vivax*, 0.57 for *P. ovale,* and 2.85 for *P. malariae* for each species-specific qPCRs. Using DNA from a culture, the limit of detection for *P. knowlesi* assay was $1.41 \times 10^{-5}$ng/mL of *P. knowlesi* DNA.

The calculated qPCR efficiencies are described in the Table S2.

### Analytical specificity

A limited number of samples were used to determine the analytical specificity of each assay (Fig. S2). The pan-*Plasmodium* assay was able to detect all five species. No cross-reactivity was found for specific assays (Table 2).

### Clinical validation

We then screened our collection of whole blood extracts obtained from patient suspected of malaria with our duplex assay. Out of 410 patients, 87 had malaria (median age, 40; sex ratio [M/F], 3.35) with positive BS with *P. falciparum* ($n = 59$), *P. vivax* ($n = 7$), *P. ovale* ($n = 10$), *P. malariae* ($n = 3$). No cases of *P. knowlesi* malaria were diagnosed (Table S3). A total of 190 samples were obtained from these 87 different positive patients, with 87 samples collected at D0 and 103 during post-therapeutic follow-up. The median number of samples per patient was 2 [Q1: 1 – Q3: 7].

### Analysis of negative samples on blood smears

Eight out of 323 patients with negative BS (2.5%) had positive duplex qPCR for at least one of the two plasmodial targets (*Plasmodium* spp./*P. falciparum*) with late Cq values (Table 3). These results were confirmed on another qPCR run. Two patients were diagnosed with positive thick blood smears in a malaria-endemic area (patients 184 and 191, Table 3). One patient had positive ICT and a negative BS (patient 322, Table 3).

**TABLE 2** Analytical specificity of each qPCR assay against 39 species-specific specimens[a]

| qPCR assay | Species-specific specimens tested | | | | |
| --- | --- | --- | --- | --- | --- |
| | *P. falciparum* (*n* = 18) | *P. vivax* (*n* = 7) | *P. ovale* (*n* = 10) | *P. malariae* (*n* = 3) | *P. knowlesi* (*n* = 1) |
| *Plasmodium* spp. | 100 | 100 | 100 | 100 | 100 |
| *P. falciparum* | 100 | 0 | 0 | 0 | 0 |
| *P. vivax* | 0 | 100 | 0 | 0 | 0 |
| *P. ovale* | 0 | 0 | 100 | 0 | 0 |
| *P. malariae* | 0 | 0 | 0 | 100 | 0 |
| *P. knowlesi* | 0 | 0 | 0 | 0 | 100 |

[a]Expressed as % of specimens detected.

**TABLE 3** Patients with negative blood smear examination and positive qPCR[a]

| No. patient | *Plasmodium* spp. Cq[b] | *P. falciparum* Cq | Internal control | Clinical informations |
|---|---|---|---|---|
| 167 | 45 | 42.22 | No inhibitors | Fever for 1 week after returning from Malaysia. Non-bloody diarrhea, abdominal pain, nausea, and dry cough |
| 182 | 45 | Negative | No inhibitors | Last trip to Congo in 2018 |
| 184 | 40.67 | 37.09 | No inhibitors | Fever and chest pain on return from a 45-day stay in Ivory Coast without prophylaxis. Positive thick drop in Ivory Coast |
| 186 | 41.59 | 37.89 | No inhibitors | Return from Cameroon |
| 188 | Negative | 40.04 | No inhibitors | Fever, diarrhea, abdominal pain, and nausea on return from a 3-day stay in Ghana |
| 191 | 42.67 | 39.04 | No inhibitors | Hospitalized in intensive care unit for altered general condition, confusional syndrome, vomiting, diarrhea, and fever. Anemia, thrombocytopenia, and acute renal failure. |
| | | | | Diagnosis and treatment of malaria based on two successive positive thick drops in Ivory Coast |
| 243 | 45 | Negative | No inhibitors | No clinical informations |
| 322 | 40.52 | 36.03 | No inhibitors | Probable treated malaria (positive *P. falciparum* ICT) |

[a]Cq, quantification cycle.
[b]A positive amplification > 45 cycles was considered positive at 45 cycles.

Two patients were hospitalized for fever and digestive disorders upon returning from malaria-endemic area (patients 167 and 188, Table 3). Three patients had few epidemiological and clinical informations (patients 182, 186, and 243, Table 3).

## Analysis on specimens obtained at diagnosis with a single species

All patients who were diagnosed by BS were positive with pan-*Plasmodium* qPCR and species-specific qPCRs (Table 4), leading a sensitivity of 100%. The sensitivity of the ICT was 89.4%, 100%, 30%, and 33.3% for the diagnosis of *P. falciparum*, *P. vivax*, *P. ovale,* and *P. malariae*, respectively (Table 4).

Using the pan-*Plasmodium* assay, the parasite load was higher for *P. falciparum* than for the other species (Cq, 26.7 for *P. falciparum* vs 28.1 for *P. vivax*, 29.4 for *P. ovale*, 29.1 for *P. malariae*, Fig. 2a; Table S4). This was also confirmed when the specific assays were used (Fig. 2b).

The correlation between the pan-*Plasmodium* and species-specific assays gave adjusted-$R^2$ between 0.71 and 0.99 (Fig. 3). When compared quantification as log of number of copies to the parasitaemia obtained by microscopy, we observed a good correlation between 13% (3,692,745 copies/µL) and less than 0.01% (85.7 copies/µL) parasitaemia (adjusted-$R^2$ = 0.8). Below 0.01% parasitaemia, qPCR assay has a wide range of quantification from 85.7 to 15,052 copies/µL (Fig. 4).

Comparing with the identification obtained by the French National Reference Center of malaria, the *P. ovale* qPCR assay was able to amplify *P. ovale wallikeri* (*n* = 1) and *P. ovale curtisi* (*n* = 4) infections.

**TABLE 4** Positivity rate for single species malaria diagnosis with BS, ICT, and qPCR

| Species | Positive BS *n* (%) | Positive ICT *n* (%) | Positive qPCR *n* (%) |
|---|---|---|---|
| *P. falciparum* | 59 (89.4) | 59 (89.4) | 66 (100) |
| *P. vivax* | 7 (100) | 7 (100) | 7 (100) |
| *P. ovale* | 10 (100) | 3 (30) | 10 (100) |
| *P. malariae* | 3 (100) | 1 (33.3) | 3 (100) |

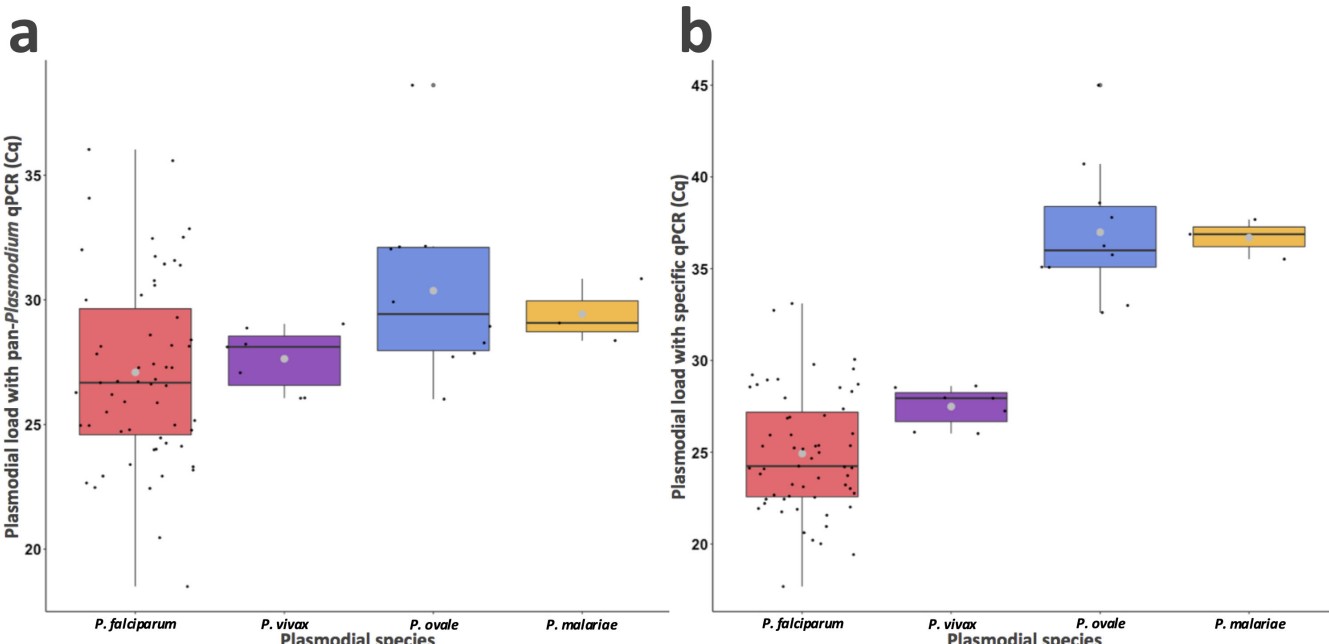

**FIG 2** Comparison of plasmodial load for *P. falciparum*, *P. vivax*, *P. ovale,* and *P. malariae* with pan-*Plasmodium* or specific-species qPCRs at the diagnosis of a single species malaria. (a) Plasmodial load with pan-*Plasmodium* qPCR, (b) plasmodial load with specific-species qPCRs. Boxplot represents median (black line) and interquartile range (IQR = Q3 − Q1) between 25th percentile (Q1) and 75th percentile (Q3). Black points represent each value of Cq, and large gray points represent the Cq average. Cq, quantification cycle.

## Analysis of patients with mixed *Plasmodium* species infection

Cases of malaria involving two different species were detected by standard methods (BS or ICT), by the qPCR used by the French National Reference Center of malaria, or during the study with our qPCR (*n* = 8). We found five patients (5.7%) harboring mixed *Plasmodium* species infection with our qPCR assays. Among them, only one mixed *Plasmodium* species infection (20%) was detected by BS. For mixed *Plasmodium* species infection involving *P. falciparum*, the parasite load of *P. falciparum* was always higher than other plasmodial species. In addition, for one mixed *Plasmodium* species infection with *P. falciparum* and *P. malariae* suspected microscopically (patient 404, Table 5), *P. malariae* qPCR assay was negative and only positive for *P. falciparum* (result confirmed by French National Reference Center of malaria). Two coinfections identified by the French National Reference Center of malaria were not found with our qPCR (patient 385 and 405, Table 5).

ICT was negative in 3/8 (37.5%) cases harboring mixed *Plasmodium* species infection by qPCR (patient 398, 403, and 405, Table 5). One sample was positive for *P. falciparum* antigen but only positive for *P. vivax* by our qPCR. This sample was positive for *P. vivax* and *P. falciparum* by National Reference Center qPCR (patient 385, Table 5). A positive sample for both antigens with ICT was confirmed with a positive qPCR for *P. falciparum* (patient 404, Table 5). Two samples positive for *P. falciparum* and *P. malariae* by qPCR were only positive for common plasmodial antigen with ICT (patient 406 and 407, Table 5). Only one sample was consistent with all techniques (patient 408, Table 5).

## Analysis on follow-up specimens from single species malaria cases

Of the 170 initial and follow-up malaria samples involving a single species, 145 (85.3%) were positive with pan-*Plasmodium* qPCR, 110 for *P. falciparum*, 13 for *P. vivax*, 17 for *P. ovale,* and 5 for *P. malariae.* Among these 145 samples, 72 (49.7%) had a parasitaemia greater than or equal to 0.01% with a copy number between 880 and 3,692,744 copies/µL. In addition, 9 (6.2%) samples had only *P. falciparum* gametocytes on the BS with copy numbers ranged from 14.9 to 1,054 copies/µL. Finally, 47 (32.4%) follow-up

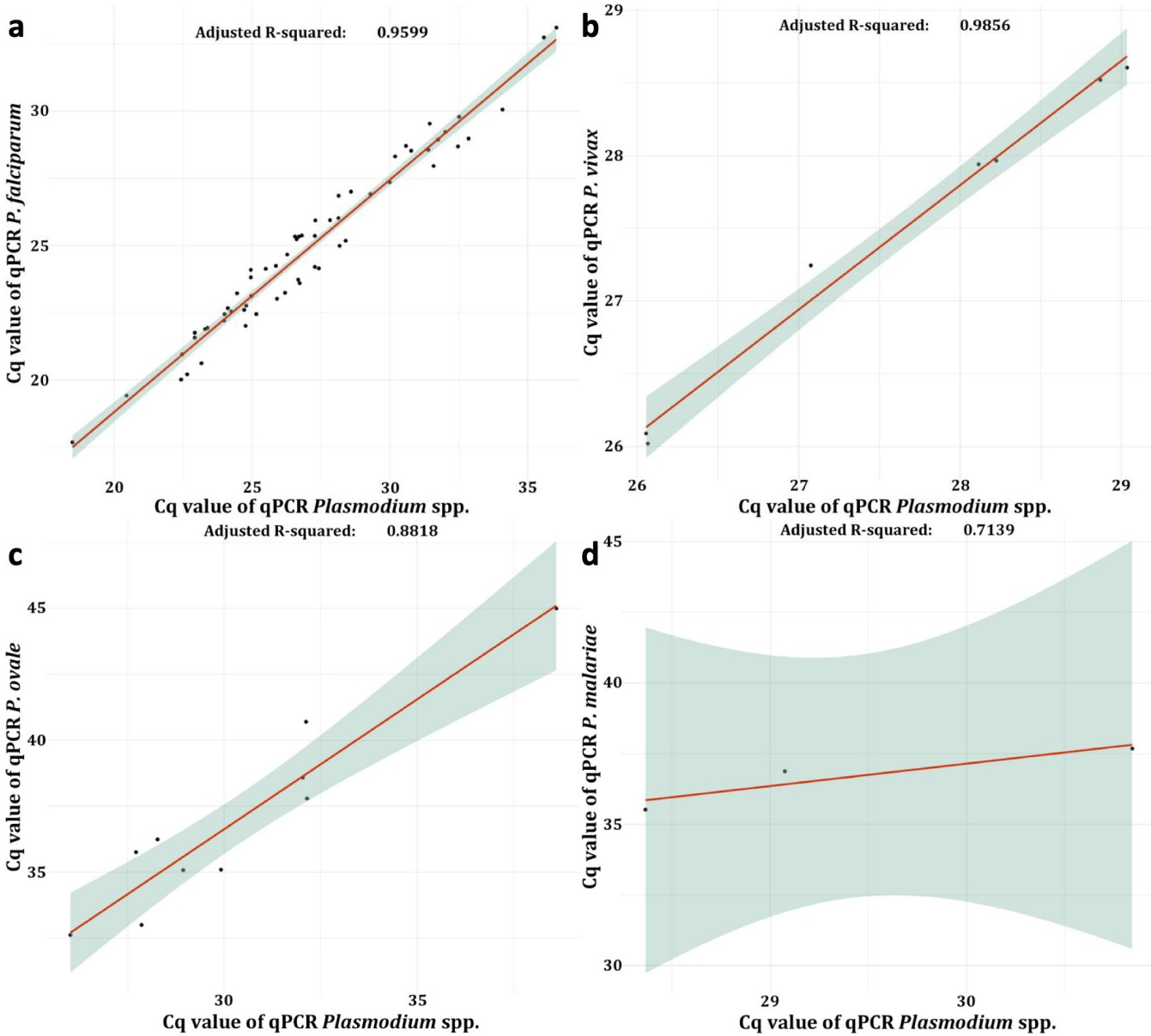

**FIG 3** Correlation between qPCR *Plasmodium* spp. and specific-species qPCRs at the single species malaria diagnosis. (a) *P. falciparum* (*n* = 59), (b) *P. vivax* (*n* = 7), (c) *P. ovale* (*n* = 10), and (d) *P. malariae* (*n* = 3). Scatterplot associated with linear trend allows to study the relation between Cq values obtained by pan-*Plasmodium* and specific-species qPCR for each plasmodial species. Cq, quantification cycle.

specimens were negative on BS and detectable with pan-*Plasmodium* qPCR with low copy numbers ranging from 2.89 to 175 copies/µL (Fig. 5). Two relapse episodes of *P. vivax* and *P. ovale* were identified. For those cases, the duplex qPCR was positive at D28, D65, and D95 for *P. vivax*, and at D56 and D332 for *P. ovale* after primary malaria infection. Each episode was treated except for the relapse episode at D28 of the primary *P. vivax* infection, which was identified with our qPCR but negative by BS.

## Analysis on follow-up specimens from *P. falciparum* cases

We then analyzed pan-*Plasmodium* qPCR results in cases for which follow-up samples under treatment were obtained (*n* = 30 patients and 85 samples). We obtained 30, 23, 18, 7, and 7 patients with a follow up at D0, D2–D4, D5–D7, D8–D14, D15–D30, respectively.

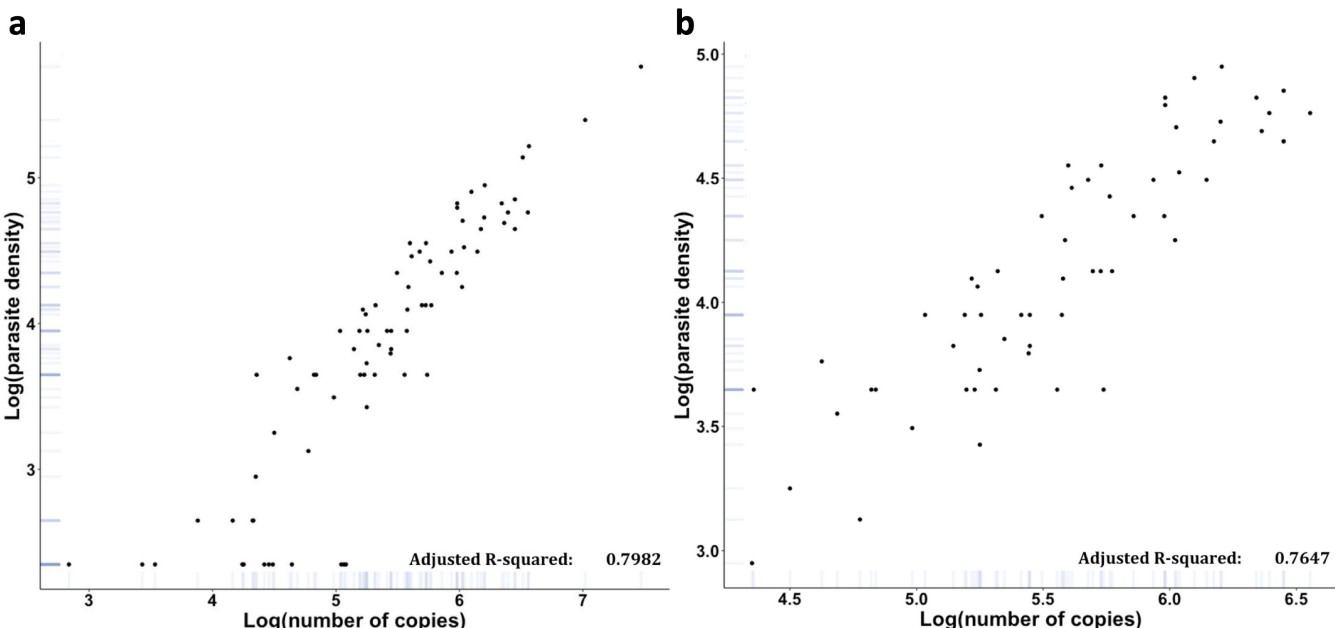

**FIG 4** Correlation between parasite density estimated by BS and number of copies with *Plasmodium* spp. qPCR at the single species malaria diagnosis for all species. (a) All parasitaemia, (b) focus on parasitaemia between more than 0.01 and less than 3%. Scatterplot allows to study the relation between a number of copies obtained by pan-*Plasmodium* qPCR and the parasite density obtained by BS. Parasite density was expressed as parasites/µL. Parasitaemia below 0.01% or the presence of gametocytes alone was represented with a value of 0.005% or 222.5 parasites/µL.

Most of the patients were treated with artemisinin-based combination therapies for 3 days. A 3.1 log decrease of the *P. falciparum* load was observed between D0 and D2–D4 specimens. The median *P. falciparum* load was 67,234 copies/µL at D0 and 53.4 copies/µL at D2–D4 (Fig. 6). Among the samples collected between D2 and D30 after the *P. falciparum* malaria infection, 95.7% and 57.1% were still positive at D2–D4 and D15–D30 follow-up, respectively. This clearance of plasmodial DNA was higher for non-*P. falciparum* species than for *P. falciparum* (Fig. S3).

## DISCUSSION

In this study, we developed a new duplex qPCR allowing the detection of both *Plasmodium* spp. and *P. falciparum*, as well as specific qPCRs for all human plasmodial species. We obtained a limit of detection of 1 copy/µL for *Plasmodium* spp. qPCR, matching with the qPCR design by Kamau et al. (16). No cross-reactivity was found for all species-specific assays. Clinical validation of these different qPCRs on a cohort of 410 patients demonstrated 100% sensitivity for the detection of all five plasmodial species, including *P. ovale wallikeri* and *P. ovale curtisi*, at the time of single-species malaria diagnosis. As described in the literature, *P. falciparum* was the species most frequently found in imported malaria cases in France (17). Analysis of negative samples by conventional microscopy methods identified 8 patients out of 323 (2.5%) with a positive amplification (Table S5). Two of these samples were a persistent circulating DNA after antimalarial treatment (patients 184 and 191, Table 3) (18). Due to the high sensitivity of qPCR, the detection of plasmodial DNA in the other 6 samples may be a detection of sub-microscopic malaria (patients 167, 186, 188, and 243, Table 3) (19), a *P. ovale/P. vivax* relapse (patient 182, Table 3), or malaria probably treated in a malaria-endemic area (patient 322, Table 3) (20). Amplifications with *P. falciparum*-specific qPCR were earlier than by *Plasmodium* spp. qPCR. Genus-targeted tests are often less sensitive than species-specific assays (21). One patient had a positive *P. falciparum*-specific qPCR and a negative *Plasmodium* spp. qPCR (patient 188, Table 3). This discrepancy may be due to a low quantity of genomic target with random amplification responding to Poisson's law, or a

**TABLE 5** Mixed *Plasmodium* species infection identified by conventional diagnostic methods and/or molecular biology[a]

| No. patient | BS | ICT | *Plasmodium* spp. Cq | *P. falciparum* Cq | *P. vivax* Cq | *P. ovale* Cq | *P. malariae* Cq[b] | *P. knowlesi* Cq | qPCR results obtained by the French National Reference Center of malaria |
|---|---|---|---|---|---|---|---|---|---|
| 385 | *P. vivax* | T1 | 34.2 | Negative | 33.2 | Negative | Negative | Negative | *P. vivax* + *P. falciparum* |
| 398 | *P. ovale* | Negative | 31.6 | 34 | Negative | 40 | Negative | Negative | *P. falciparum* + *P. ovale curtisi* |
| 403 | *P. malariae* | Negative | 33 | 35.9 | Negative | Negative | 40 | Negative | NA |
| 404 | *P. falciparum* & *P. malariae* | T1 +T2 | 31.4 | 29.5 | Negative | Negative | Negative | Negative | *P. falciparum* |
| 405 | *P. malariae* | Negative | 31.6 | Negative | Negative | Negative | 39.6 | Negative | *P. ovale* + *P. malariae* |
| 406 | *P. malariae* | T2 | 29.5 | 32.5 | Negative | Negative | 36.2 | Negative | *P. malariae* |
| 407 | *P. malariae* | T2 | 28.4 | 31.9 | Negative | Negative | 35.5 | Negative | *P. falciparum* + *P. malariae* |
| 408 | *P. falciparum* & *P. malariae* | T1 +T2 | 29.3 | 27.7 | Negative | Negative | 45 | Negative | *P. falciparum* + *P. malariae* |

[a]T1 and T2 were the *P. falciparum*-specific antigen (HRP2) and the common antigen of all plasmodial species, respectively. NA, not applicable.
[b]A positive amplification > 45 cycles was considered positive at 45 cycles.

better efficiency of specific-*P. falciparum* qPCR. Despite the presence of symptoms suggestive malaria, travel to malaria-endemic areas, and qPCR duplex results verifications, false-positive results cannot be formally excluded for these patients (22). A study carried out in Colombia showed a global malaria prevalence of 0.3% by microscopy, compared with an estimated 9.7% by qPCR (23). These results support the value of robust and sensitive molecular tools for the early diagnosis of sub-microscopic malaria cases (24). A French study carried out by the National Reference Center of malaria showed a lower sensitivity of commercial assays (BIOSYNEX Ampliquick Malaria and Bio-Evolution *Plasmodium* Typage assays) as compared to LAMP-based assay Alethia Malaria and in-house TaqMan qPCR, particularly in negative microscopic malaria cases (25).

The interest of this new diagnostic tool is based in the use of a duplex qPCR able to detect a pan-plasmodial target, a *P. falciparum*-specific target, and a viral internal control. In contrast to the LAMP method, we are able to detect all human plasmodial species and to quantify plasmodial number of copies (26). Despite its cost, qPCR is the most sensitive method for malaria diagnosis. The development of this new tool is based on the use of a primer and probe common to all plasmodial species in order to reduce reagent costs (27).

Using microscopic examination as the gold standard for malaria diagnosis, duplex qPCR and species-specific qPCRs have a diagnostic sensitivity of 100% for cases of monoparasitism (Table S6). The lack of sensitivity of ICT for the diagnosis of *P. ovale* and *P. malariae* infections was confirmed in our study, with a sensitivity of 30% and 33.3%, respectively (27). For *P. ovale* infections, we detected both *P. ovale wallikeri* and *P. ovale curtisi* infections as identified by the French National Reference Center of malaria (28). Despite the detection of all cases of *P. ovale* and *P. malariae* monoparasitism, we observed later Cq with species-specific qPCRs, compared with those for *P. falciparum* and *P. vivax*. Parasite density of asexual forms was observed to be higher for *P. falciparum* infections compared with those involving *P. ovale* and *P. malariae* (29). The use of degenerate primers or probes could be tested to obtain earlier Cq (30). Although there are very few descriptions of imported *P. knowlesi* malaria in France (6 cases since 2010, data of French National Reference Center of malaria), we have developed a specific qPCR for this species (31).

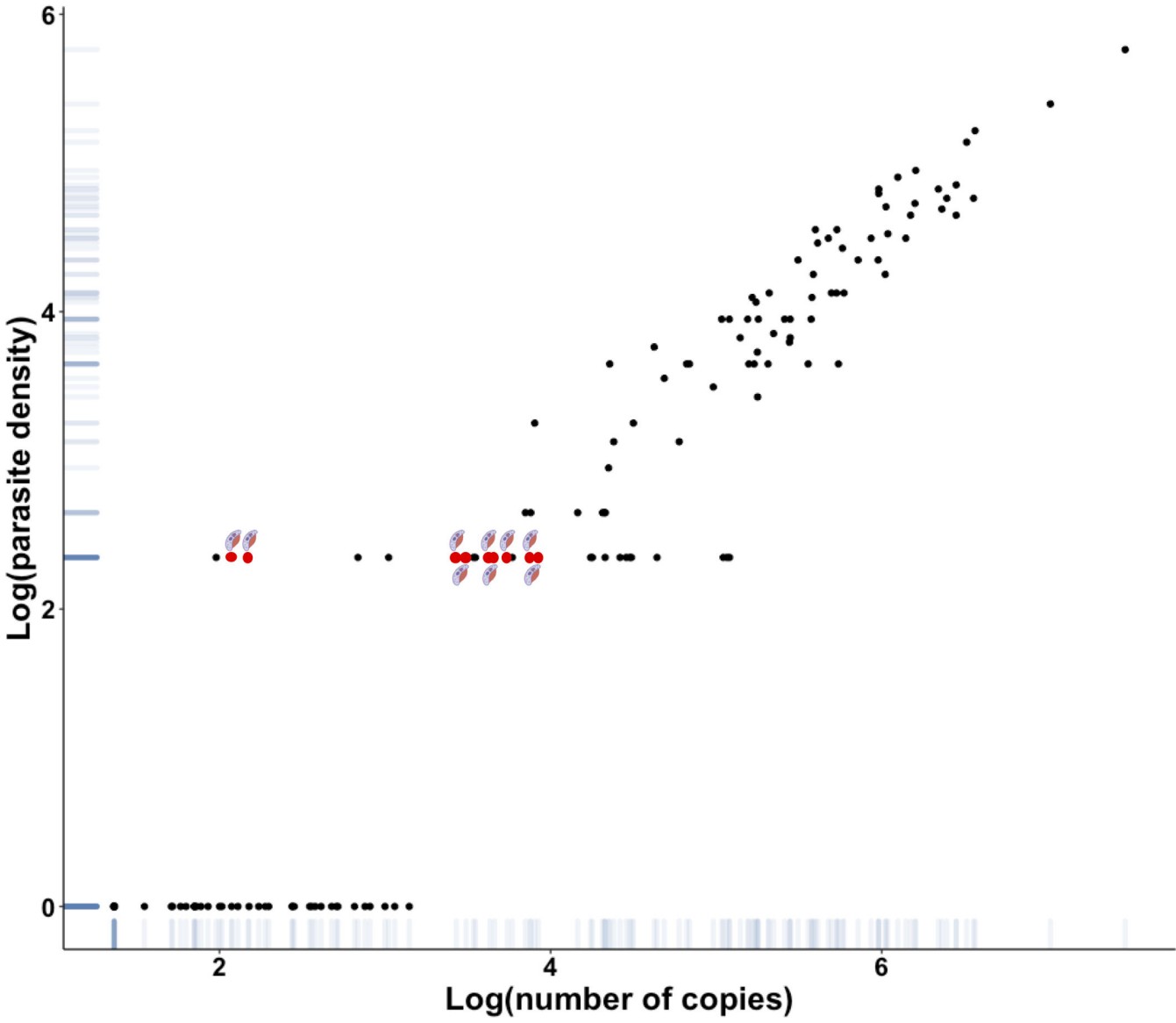

**FIG 5** Representation of parasite density estimated by BS vs number of copies with *Plasmodium* spp. qPCR on all species at diagnosis and follow-up for single species malaria. Scatterplot allows to study the relation between number of copies obtained by pan-*Plasmodium* qPCR and the parasite density obtained by BS. A positive amplification >45 cycles was considered positive at 45 cycles or 23.1 copies. Parasitaemia below 0.01% or the presence of gametocytes alone was considered with a value of 0.005% or 222.5 parasites/µL. Red points represent the presence of gametocytes alone on BS.

The use of a plasmid containing a fragment of the small 18S subunit of ribosomal DNA enabled us to assess the consistency between microscopy and qPCR for parasitaemia determination. Despite a good correlation between microscopic parasite density and a number of copies (25), we observed a heterogeneous distribution of parasite load within low parasitaemia probably due to a combination of sequestration and stochastic effects of low copy number DNA (32). The study of Ballard et al. confirms the value of using the 18S rRNA qPCR, in intermediate parasitaemia values, to assess parasite load during *P. falciparum* malaria (33).

Among the 87 microscopy-positive patients, we identified 5 cases (5.7%) of mixed *Plasmodium* species infection by qPCR. It is important to diagnose mixed infections involving *P. vivax* and *P. ovale* in order to initiate treatment by primaquine or tafenoquine for the prevention of relapse episodes (34). When comparing our qPCR results with those of the French National Reference Center of malaria, we observed three discrepancies

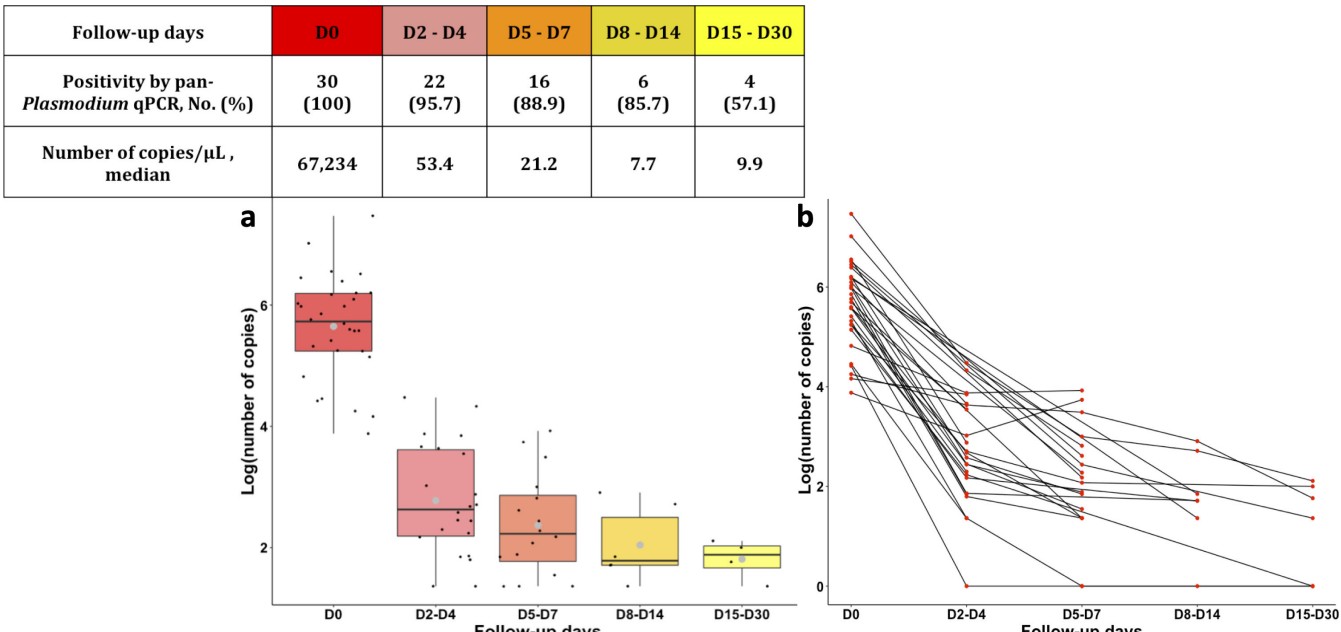

| Follow-up days | D0 | D2 - D4 | D5 - D7 | D8 - D14 | D15 - D30 |
|---|---|---|---|---|---|
| Positivity by pan-*Plasmodium* qPCR, No. (%) | 30 (100) | 22 (95.7) | 16 (88.9) | 6 (85.7) | 4 (57.1) |
| Number of copies/µL, median | 67,234 | 53.4 | 21.2 | 7.7 | 9.9 |

**FIG 6** Persistence of circulating *P. falciparum* DNA during post-treatment follow-up with pan-*Plasmodium* qPCR. (A) Boxplot represents median (black line) and interquartile range (IQR = Q3 − Q1) between 25th percentile (Q1) and 75th percentile (Q3) of the *P. falciparum* copy number decrease as a function of post-therapy follow-up. Black points represent each value of *P. falciparum* copy number, and large gray points represent the *P. falciparum* copy number average. (B) Connected scatterplot represents the *P. falciparum* copy number decrease for each patient.

(patient 385, 405, and 406, Table 5). It would be interesting to conduct a comparative study between the qPCR developed in our study and the qPCR used by the French National Reference Center of malaria (25). The results of the mixed *Plasmodium* species infection cases involving *P. falciparum* are consistent with those of the French National Reference Center of malaria. In 37.5% of mixed *Plasmodium* species infection cases, the ICT was negative, confirming the lack of sensitivity of immunochromatographic tests (35).

To date, biological assessment of plasmodial clearance is based on the use of microscopy. ICT and qPCR are not recommended because of parasite persistence after antimalarial treatment. In our study, 145 samples from single species malaria cases (85.3%) were positive by qPCR at diagnosis and during post-treatment follow-up, with a sharp decrease in parasite load between D0 and D3. The French retrospective study by Kamaliddin et al. showed that rapid diagnostic test and qPCR remained positive, respectively, in 51% and 10%–12% of cases, 28 days after treatment of imported *P. falciparum* malaria (36). Despite the small size of our study, we detected persistent *P. falciparum* DNA in 57.1% of samples between 15 and 30 days after treatment. The difference in parasite DNA clearance observed by qPCR may be explained by differences in premunition status between patients (37). qPCR positivity after anti-malarial treatment can be explained by better sensitivity of qPCR compared to conventional methods, elimination of parasite DNA after parasite death, and sub-microscopic asexual blood-stage parasite or gametocytes persistence (37–39). Indeed, we found positive qPCR for 9 samples (6.2%) in which gametocytes were found on microscopic examination without other blood-stage parasite forms, with a plasmodial load of between 119.2 and 8,432 DNA copies (Fig. 5). A future perspective of our study would be to perform specific gametocyte detection by RT-qPCR (Reverse Transcriptase-qPCR) on our total nucleic acid extracts. The prevalence of gametocytes estimated by Mwingira et al. was determined by microscopy and pfs25-specific RT-qPCR in 226 patient samples (40). The results showed a gametocyte carriage in the study population of 10.6% by RT-qPCR and 1.2% by microscopy. These results demonstrate that the use of qPCR is interesting for

the detection of sub-microscopic gametocytemia in patients who may be the reservoir of malaria. We have shown that *P. falciparum* load decays less rapidly than other plasmodial species (Fig. 6). The study of post-therapeutic plasmodial load decay by Lo et al. showed faster clearance for *P. vivax* than for *P. falciparum* (18). This may be explained by the extended circulation of *P. falciparum* gametocytes, compared with that of other species (41). This tool could be used to study the decay of post-therapeutic parasite load in non-endemic areas. Indeed, it would be interesting to study its relevance to the persistence of circulating DNA in a cohort of patients infected with antimalarial drug-resistant mutants (18). Two *P. vivax* and *P. ovale* relapses were identified. In these cases, duplex qPCR was positive at D28, D65, D67, and D95 for *P. vivax* and at D56, D59, and D332 for *P. ovale* after initial malaria diagnosis. In comparison with these qPCR results, microscopic examination was negative at D28 and D67 for *P. vivax* and at D59 for *P. ovale*. No information was reported on treatment by primaquine or tafenoquine after the primary episode. The absence of such treatment may explain the later relapse (42). No treatment failure occurred in patients whose samples were included in this study, confirmed by the absence of resistance tested by the French National Reference Center of malaria.

Our study regarding these new *Plasmodium* qPCR assays has some limitations. Indeed, we had a limited sample size for non-*P. falciparum* species. Because of limited number of imported *P. knowlesi* malaria cases in France, we were not able to test clinical samples to study properly the performance of the pan-*Plasmodium* and the specific *P. knowlesi* qPCR assays. Finally, we lacked clinical information to better describe and explain positive qPCRs in patients with negative results by conventional techniques. In addition, these samples were not checked with another molecular biology assay.

A prospective study is needed to fully evaluate this new tool, both at diagnosis and during post-therapeutic follow-up of malaria. In fact, we propose to use this new tool alongside conventional methods, using *Plasmodium* spp./*P. falciparum*/internal control qPCR in a suspected malaria of patient returning from an endemic area. If both

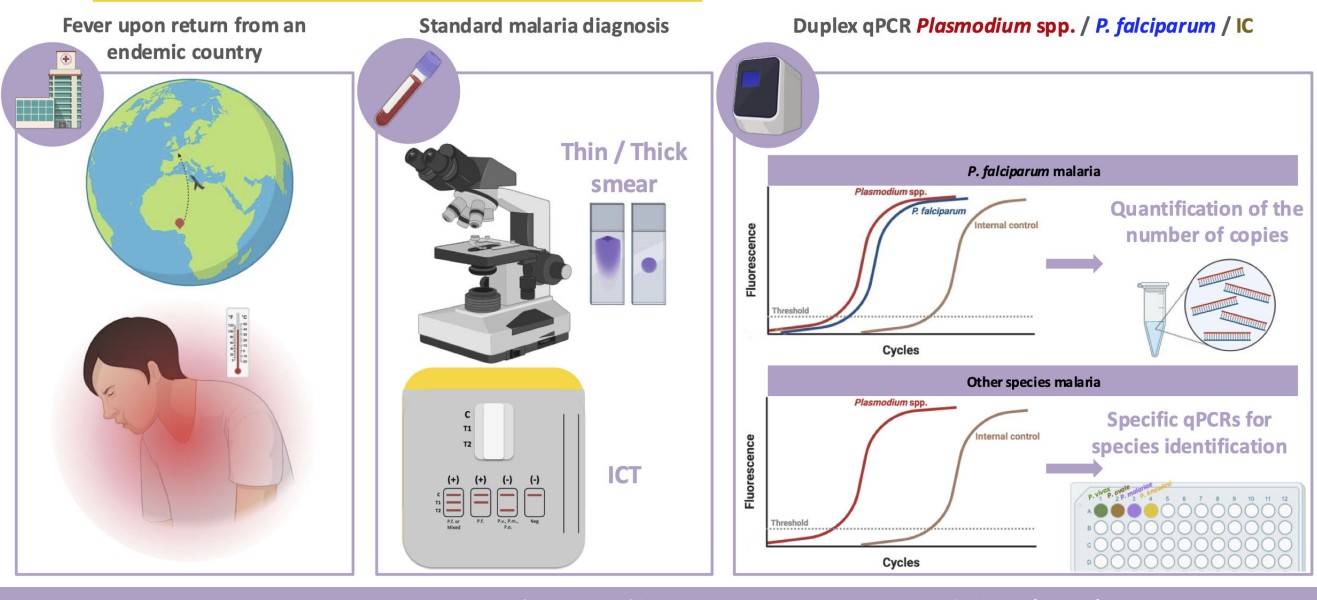

**FIG 7** (graphical abstract). Integration of *Plasmodium* qPCR assay in the biological diagnosis of malaria.

plasmodial targets of duplex qPCR are positive, we can quantify the number of copies. In the case of single *Plasmodium* spp. target positivity, suggesting malaria caused by a species other than *P. falciparum*, we will use specific qPCRs to identify the involved species (Fig. 7). Finally, we will develop the automation and miniaturization of these tests to enable DNA extraction and amplification in less than two hours, with the aim of establishing a reliable and rapid diagnosis.

## ACKNOWLEDGMENTS

We thank the technical staff of the Mycology-Parasitology laboratory at Hospital Saint-Louis, Paris, France, for handling specimens routinely, in particular, Élodie Da Silva, Thierry Pautet, Dieyenaba Siby-Diakite, Sarah Seng, and Julie Bui. Thanks to Mahdi Ouafi for proofreading the manuscript.

C.C. performed the experiments, collected the data, and wrote the original draft; C.C. and A.A. interpreted and analyzed the data, performed statistical analysis, edited and reviewed the draft; S.Ha. and E.DS. contributed to routine analysis; S.Ha., T.G.F., S.D., E.DS., B.D., S.Ho., and V.J. contributed to review and edit the manuscript; A.A. supervised the study and all reviewed the final version of manuscript.

## AUTHOR AFFILIATIONS

[1]Laboratoire de Parasitologie-Mycologie, Hôpital Saint-Louis, Assistance Publique-Hôpitaux de Paris, Paris, France

[2]Laboratoire de Parasitologie-Mycologie, INSERM U1285, Unité de Glycobiologie Structurale et Fonctionnelle (CNRS UMR 8576), Centre Hospitalier Universitaire de Lille, Université de Lille, Lille, France

[3]Institut Pasteur, Université Paris Cité, National Reference Center for Invasive Mycoses and Antifungals, Translational Mycology research group, Mycology Department, Paris, France

[4]Service des maladies infectieuses, Hôpital Saint-Louis, Assistance Publique-Hôpitaux de Paris, Paris, France

[5]Centre National de Référence du Paludisme, Laboratoire de Parasitologie-Mycologie, Hôpital Bichat-Claude Bernard, Paris, France

[6]Université Paris Cité, IRD, MERIT, Paris, France

## AUTHOR ORCIDs

Camille Cordier http://orcid.org/0009-0005-8923-8764
Théo Ghelfenstein-Ferreira http://orcid.org/0000-0002-7784-4434
Sarah Dellière http://orcid.org/0000-0003-3821-1762
Valentin Joste http://orcid.org/0000-0002-9387-5388
Alexandre Alanio http://orcid.org/0000-0001-9726-3082

## AUTHOR CONTRIBUTIONS

Camille Cordier, Conceptualization, Data curation, Formal analysis, Funding acquisition, Investigation, Methodology, Project administration, Software, Validation, Visualization, Writing – original draft, Writing – review and editing | Samia Hamane, Data curation, Investigation, Methodology, Project administration, Resources, Validation, Visualization, Writing – review and editing | Théo Ghelfenstein-Ferreira, Validation, Visualization, Writing – review and editing | Sarah Dellière, Investigation, Methodology, Validation, Visualization, Writing – review and editing | Élodie Da Silva, Investigation, Methodology, Validation, Visualization, Writing – review and editing | Blandine Denis, Validation, Visualization, Writing – review and editing | Sandrine Houzé, Resources, Validation, Visualization, Writing – review and editing | Valentin Joste, Resources, Validation, Visualization, Writing – review and editing | Alexandre Alanio, Conceptualization, Data curation, Formal analysis, Funding acquisition, Investigation, Methodology, Project administration, Resources, Supervision, Validation, Visualization, Writing – original draft, Writing – review and editing

## DATA AVAILABILITY

Data will be made available upon request.

## ETHICS APPROVAL

All specimens have been tested as part of the routine diagnostic procedure in Saint-Louis Hospital, Paris, France (current care). This non-interventional study on leftover specimens did not require approval of an ethics committee according to the French Health Public Law (CSP Art L1121-1.1). Patient were anonymized and basic data have been recorded and listed retrospectively. None of the results generated in this study impacted the clinical management.

## ADDITIONAL FILES

The following material is available online.

### Supplemental Material

**Supplemental tables and figures (Spectrum01622-24-s0001.docx).** Tables S1 to S4; Fig. S1 to S3.
**Table S5 (Spectrum01622-24-s0002.xlsx).** qPCR duplex results for negative patients by BS.
**Table S6 (Spectrum01622-24-s0003.xlsx).** qPCR duplex and specific-species qPCRs results for positive patients by BS.

### Open Peer Review

**PEER REVIEW HISTORY (review-history.pdf).** An accounting of the reviewer comments and feedback.

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
