## [Reviewer comments · Microbiology Spectrum]

Microbiology Spectrum

Implementation and validation of a new qPCR assay to detect imported human *Plasmodium* species.

Camille Cordier, Samia Hamane, Théo Ghelfenstein-Ferreira, Sarah Dellière, Elodie Da Silva, Blandine Denis, Sandrine Houzé, Valentin Joste, and Alexandre Alanio

Corresponding Author(s): Camille Cordier, Centre Hospitalier Universitaire de Lille

Review Timeline:

Submission Date:	July 3, 2024
Editorial Decision:	September 9, 2024
Revision Received:	November 6, 2024
Editorial Decision:	November 7, 2024
Revision Received:	November 8, 2024
Accepted:	November 11, 2024

Editor: Wendy Szymczak

Reviewer(s): Disclosure of reviewer identity is with reference to reviewer comments included in decision letter(s). The following individuals involved in review of your submission have agreed to reveal their identity: Abhinav Sinha (Reviewer #2)

Transaction Report:

DOI: <https://doi.org/10.1128/spectrum.01622-24>

Re: Spectrum01622-24 (**Implementation and validation of a new qPCR assay to detect imported human *Plasmodium* species.**)

Dear Dr. Camille Cordier:

Thank you for the privilege of reviewing your work. Below you will find my comments, instructions from the Spectrum editorial office, and the reviewer comments.

Revision Guidelines

Sincerely,
Wendy Szymczak
Editor
Microbiology Spectrum

Reviewer #1 (Comments for the Author):

This is a long manuscript, but it describes a rather simple study, characterizing a new qPCR protocol to identify malaria parasitemia. Strengths of the study are meticulous attention to detail and the availability of a large number of samples for study from patients diagnosed with malaria in France. However, the manuscript has a number of problems with communication and also offers some misleading statements. Also, it is unclear just what advantages are gained with this new protocol compared to

others that have previously been reported or marketed. Lastly, the manuscript offers a great deal of detail, and it could be shortened considerably without compromising its most important messages. Some specific concerns are as follows:

- 1) Line 36. The term "biparasitism" is odd and confusing in English. This should be changed to "mixed Plasmodium species infection" or a similar term here and elsewhere in the MS.
- 2) Line 39. The meaning of "epidemiological and clinical arguments of malaria" is unclear. The need for brevity in the abstract is appreciated, but nonetheless this needs clarification.
- 3) Line 39. The last sentence of Results does not offer results, but rather an interpretation: "Plasmodium spp. qPCR is a useful tool for monitoring post-therapeutic parasite DNA decrease." This should be replaced by a statement describing the results.
- 4) Abstract Interpretation. This section describes results not mentioned earlier in the abstract ("allowing the detection of all human plasmodial species"). The Results section should better summarize the most important results. The Interpretation section should not offer results not described earlier, but rather should interpret the relevance of results described above.
- 5) Line 47. "High decrease" is a confusing terminology. Rewriting is needed.
- 6) Line 48. The numbers of cases and deaths were not "recorded". Rather, these are estimates based on complex modeling. Also, the authors should include the most recent available WHO estimates, for 2022, from the 2023 World Malaria Report, which was published in late 2023.
- 7) Line 50. Direct examination of what?
- 8) Line 65. It should be noted that these numbers are based on estimates; they are not hard statements of incidence. As above, case numbers were based on estimates, not reports.
- 9) Lines 76 and 114. The mention of trophozoites and schizonts is misleading. The vast majority of cases are *P. falciparum*, in which trophozoites and schizonts do not circulate, and diagnosis is based on identification of ring forms. Rings are sometimes referred to as "ring trophozoites", causing some confusion, but this sentence would be clearer without mention of noncirculating stages of the parasites.
- 10) Line 93. Again, biparasitism is not standard English. This correction is also needed elsewhere in the MS.
- 11) Line 97. Circulant does not have the intended meaning in English. Circulating would be appropriate here and elsewhere in the MS.
- 12) Line 108. Although this is not an epidemiology study, more epidemiological information is needed. Where were these patients studied? In what countries was malaria presumed to have been acquired? Were all patients symptomatic with symptoms and signs suggestive of malaria? Were all patients studied before administration of antimalarial drugs? Had any of them received chemoprophylactic therapy during travel?
- 13) Line 116. Who performed microscopy? Were slides double read and if so, how were discordant readings adjudicated?
- 14) Line 121. The study benefitted from a qPCR kit capable of identifying all species that is already available in kit form (Plasmodium Typage (Bio-Evolution, Bussy-Saint-Martin, France) real-time qPCR kit). The specific advance represented by the new qPCR methodology introduced with this manuscript needs to be more clearly elaborated, with comparison to prior available methods including this kit and published methods.
- 15) Figure 1. This figure nicely demonstrates phylogenetic relationships, but it is not clear why multiple clones are shown for multiple species, but only for *P. falciparum* these are labelled as two separate lineages. The reason for this should be explained. Also, the basis of the numbers provided in the tree should be explained so that readers can easily search for the sequence data.
- 16) Line 186. Change "parasitiform forms" to "parasites".
- 17) Line 208. The authors emphasize the two related species of *P. ovale* elsewhere in the MS, but lump these together here. If they wish to emphasize diagnosis of both species they need to distinguish the two throughout the MS. Alternatively, since very few *P. ovale* parasites were studied, assertions that the protocol can distinguish the two species should be modified.
- 18) Line 244. "The limit of detection....was detectable"? Correction is needed.
- 19) Line 247. It is important that, for *P. ovale*, the two sub-species were not distinguished and, for *P. knowlesi*, only a culture isolate could be studied.
- 20) Line 263. The text refers to "all 6 species", but in fact results for only 5 species are shown in Table 2 (with results for the two *P. ovale* sub-species lumped together).
- 21) Line 283 "Epidemiological or clinical data, available for 7 of them, were consistent with the detection of low plasmodial DNA load in blood (Table 3)." This sentence is unconvincing. Evidence for malaria in these patients is very limited. The authors should not claim here or in the abstract (lines 38-39) that there was convincing support for the diagnosis of malaria based on epidemiological information. Also, in Table 3, the meaning of "Two thick drops positive in Ivory Coast with malaria treatment." is unclear. Overall, it is possible that the patients with low positive qPCR readings had malaria infections, but also possible that these were false positives.
- 21) Line 389. These episodes of recurrent *P. vivax*/*P. ovale* infection were presumably considered relapses, and this term, rather than "reactivation" should be used. Also, it would be helpful to know if these patients were treated with primaquine or tafenoquine for their primary infections.
- 22) Line 451. The precise meaning of "pauci-symptomatic" is unclear, and it is not known if any of the cases detailed in Table 3 represented smear negative symptomatic malaria. The Discussion should discuss potential benefits (improved sensitivity) and risks (decreased specificity) of a more sensitive assay to detect plasmodial DNA.
- 23) Line 453. As noted above (point 14) it would be helpful here to explain the specific advantages of the new qPCR system described in this manuscript compared to others that have already been reported or are sold commercially.
- 24) Line 480. The parenthetical statement is not clear; it should be replaced by a sentence explaining the key point regarding *vivax* and *ovale* infections.
- 25) Lines 488 and 497. Replace "secondary", which implies causality, with "after."

26) Line 496. The sentence beginning "The positivity of qPCR after anti-malarial treatment can be explained" is not clear and is incomplete. Likely the main explanations for persistent PCR identification after treatment are persistent parasite DNA for some days after parasite killing and circulating gametocytes, which are not killed by most therapies. Gametocytes are mentioned a few lines later, but the arguments should be consolidated into a simpler explanation.

27) Line 500. Change "parasitical" to "parasite".

28) A limitations paragraph would be helpful. This could include the following limitations, and possibly others: a) few nonfalciparum and no *P. knowlesi* clinical isolates were studied; b) clinical information for study subjects was very limited; c) the study was not equipped to identify false positive results from the qPCR assay (including clinical false positives for patients with circulating plasmodial DNA but no active infection).

Reviewer #2 (Comments for the Author):

The authors have developed a qPCR assay based on 18S rRNA gene and compared it with the outcomes of BS (considering it as a gold standard) & ICT. The LOD of the qPCR presented by the authors was 8 copies/reaction. It's not apparent what they mean by copies in this context. Are these copies of the target gene or the genome of the parasite or plasmid? Further reactions may be of any volume, so the LOD would vary accordingly. Therefore, authors should be very precise when presenting such key outcomes. Moreover, as per authors control-plasmid was detectable until a dilution of 1 copies/ μ L & using clinical samples, the limit of detection was between 0.02 & 2.85 parasites/ μ L depending on the species. How this '8 copies/reaction' came into picture is not described or discussed anywhere in results.

In the abstract, the authors mentioned five cases of biparasitism (5.7%) by qPCR and concluded qPCR better than BS and ICT. However, in supplemental table-3 they presented 8 cases of biparasitism (9.2%) by BS. It is not clear how they concluded this outcome? Further in table 5, out of the 8 cases, only 2 were having biparasitism by BS. Such outcomes seem to be very confusing throughout the manuscript.

It is unclear whether each sample has been tested for all Plasmodium species through species-specific qPCR assay or not except 38 samples (positive for a single species by BS & ICT). Because that would be must to see the cross-reactivity. Also, the qPCR outcomes were compared with BS which is very inferior (lower LOD) to qPCR. The actual results of clinical samples should be known before validating a method using those samples. Although, the authors mentioned species confirmation was done by the real-time qPCR kit (French National Reference Center) but details (for example: based on which target gene & LOD) of that kit was not given.

Nucleic Acid Extraction: the authors mentioned they added an internal control per sample. It has not been detailed further why this addition was done? Authors simply have given a statement.

Validation of qPCR: Authors stated, 333 samples from 323 patients suspected of malaria but negative by BS and ICT were screened with duplex qPCR assay. Later, under "Analysis of negative samples on blood smears" they said one was positive by ICT. Such negligence creates doubt on the presented outcomes.

In table 3, sample negative for Plasmodium spp. & positive for *P. falciparum* is not justifiable. All species-specific positive samples must be positive for Plasmodium spp. Further, logically the cq values of Plasmodium spp. should be less than that of species-specific cq values but the authors observed just opposite.

Dear Wendy Szymczak,

We thank you for the reviewing of our manuscript “Implementation and validation of a new qPCR assay to detect imported human *Plasmodium* species” to Spectrum.

Your comments have greatly improved this manuscript, clarifying some points and adding details to the text. We think that the changes made to the manuscript will enable us to publish in your journal.

Please find attached the Marked-Up Manuscript and the response to reviewers.

**We thank you for considering our revised manuscript,
Sincerely,
Dr. Camille Cordier and Pr. Alexandre Alanio.**

Response to Reviewer 1

1) Line 36. The term "biparasitism" is odd and confusing in English. This should be changed to "mixed *Plasmodium* species infection" or a similar term here and elsewhere in the MS.

Reply: We agree, the term "biparasitism" can be confusing.

Action: The term "biparasitism" has been replaced by "mixed *Plasmodium* species infection" when appropriate.

2) Line 39. The meaning of "epidemiological and clinical arguments of malaria" is unclear. The need for brevity in the abstract is appreciated, but nonetheless this needs clarification.

Reply: This sentence means that we have identified a positive duplex qPCR in patients who presented clinical symptoms on their return from a malaria-endemic area.

Action: The previous sentence has been replaced by the following: "eight out of 323 patients with negative BS hospitalized for symptoms suggestive malaria after malaria-endemic area travel or known with history of malaria had a low positive duplex qPCR" (lines 36 to 38).

3) Line 39. The last sentence of Results does not offer results, but rather an interpretation: "*Plasmodium* spp. qPCR is a useful tool for monitoring post-therapeutic parasite DNA decrease." This should be replaced by a statement describing the results.

Reply: Indeed, this sentence is more about interpretation than results.

Action: This sentence has been replaced by a following result sentence: "between day of diagnosis and post-treatment follow-up at D2-D4 of malaria, a 3.1-log *P. falciparum* load decrease was observed by qPCR" (lines 38 to 39).

4) Abstract Interpretation. This section describes results not mentioned earlier in the abstract ("allowing the detection of all human plasmodial species"). The Results section should better summarize the most important results. The Interpretation section should not offer results not described earlier, but rather should interpret the relevance of results described above.

Reply: Further details have been added in the results section to add value to the interpretation that has been carried out.

Action: The interpretation sentence has been retained, but the following sentence has been added to the results: "Seventy-nine patients diagnosed for single species malaria involving *P. falciparum*, *P. vivax*, *P. ovale* and *P. malariae* were positive with duplex and species-specific qPCRs. *P. knowlesi* qPCR detected DNA from a *P. knowlesi* culture" (lines 32 to 34).

5) Line 47. "High decrease" is a confusing terminology. Rewriting is needed.

Action: "high decrease" has been replaced by "significant drop" (line 105).

6) Line 48. The numbers of cases and deaths were not "recorded". Rather, these are estimates based on complex modeling. Also, the authors should include the most recent available WHO estimates, for 2022, from the 2023 World Malaria Report, which was published in late 2023.

Reply: Data have been updated with estimates from WHO 2023 report.

Action: "recorded" have been replaced by "estimated" (lines 106 and 136). The sentence has been modified to present data from the WHO 2023 report for the year 2022 as "despite a significant drop of incidence and mortality since 2000, 249 million cases and 608,000 deaths have been estimated in 2022, mainly in Africa" (lines 105 to 106).

7) Line 50. Direct examination of what?

Reply: This sentence lacked precision.

Action: The sentence was completed by: "microscopic examination of thin and thick smears" (lines 108 to 109).

8) Line 65. It should be noted that these numbers are based on estimates; they are not hard statements of incidence. As above, case numbers were based on estimates, not reports.

Reply: the "estimation" term has been used for this part of the text. Data has been updated.

Action: The sentence has been modified as in point 6: "despite this, 249 million cases and 608,000 deaths were estimated in 2022, mostly in Africa and among children under 5 years" (lines 136 to 137).

9) Lines 76 and 114. The mention of trophozoites and schizonts is misleading. The vast majority of cases are *P. falciparum*, in which trophozoites and schizonts do not circulate, and diagnosis is based on identification of ring forms. Rings are sometimes referred to as "ring trophozoites", causing some confusion, but this sentence would be clearer without mention of noncirculating stages of the parasites.

Reply: Ring-shaped forms correspond indeed to young *Plasmodium* trophozoites. We agree with you that schizont forms are exceptionally found in *P. falciparum* malaria.

Action:

Lines 145 and 197, the sentence "trophozoites or schizonts" has been replaced by "ring-stage trophozoites".

Lines 196 to 198, the sentence "malarial episode was defined as the presence of trophozoites and/or schizonts detected upon thick smears..." has been replaced by "malarial episode was defined as the presence of ring-stage trophozoites on blood smears...".

10) Line 93. Again, biparasitism is not standard English. This correction is also needed elsewhere in the MS.

Action: The term biparasitism has been replaced by "mixed *Plasmodium* species infection" throughout the manuscript.

11) Line 97. Circulant does not have the intended meaning in English. Circulating would be appropriate here and elsewhere in the MS.

Action: “circulant” has been replaced by “circulating” here and elsewhere in the manuscript.

12) Line 108. Although this is not an epidemiology study, more epidemiological information is needed. Where were these patients studied? In what countries was malaria presumed to have been acquired? Were all patients symptomatic with symptoms and signs suggestive of malaria? Were all patients studied before administration of antimalarial drugs? Had any of them received chemoprophylactic therapy during travel?

Reply: One of the study aims was to compare qPCR performances with basic tests used for malaria diagnosis (microscopic examination and immunochromatographic test) in clinical samples. We wanted to evaluate if we didn't miss a malarial diagnosis with our qPCRs. These patients were admitted to the emergency department of Saint-Louis hospital, Paris, France for suspicion of malaria in febrile patients returning from malaria-endemic areas. As in many situations, we do not always have information on chemoprophylaxis or administration of antimalarial treatments that occurred in endemic areas.

Action: The following sentence (lines 194 to 195) has been added “these patients were admitted to the emergency department of Saint-Louis hospital, Paris, France for malaria suspicion in febrile patients returning from malaria-endemic areas.”

13) Line 116. Who performed microscopy? Were slides double read and if so, how were discordant readings adjudicated?

Reply: Microscopic examination was performed by a laboratory technician and a biologist. Discordant readings were checked and resolved by a senior experienced medical biologist.

Action: The following sentence (lines 199 to 201) has been added “microscopic examination was performed by double reading. In case of discrepancy, a medical senior biologist resolved the case”.

14) Line 121. The study benefitted from a qPCR kit capable of identifying all species that is already available in kit form (*Plasmodium* Typage (Bio-Evolution, Bussy-Saint-Martin, France) real-time qPCR kit). The specific advance represented by the new qPCR methodology introduced with this manuscript needs to be more clearly elaborated, with comparison to prior available methods including this kit and published methods.

Reply: Due to length limitations, we choose not to establish a comparison with the published or commercial qPCR assays. To respond to this point, we have written a comparison point with other existing qPCRs in the discussion (point 23).

Action: Advantages of the new qPCR were cited (lines 690 to 718) in the discussion (see question 23).

15) Figure 1. This figure nicely demonstrates phylogenetic relationships, but it is not clear why multiple clones are shown for multiple species, but only for *P. falciparum* these are labelled as two separate lineages. The reason for this should be explained. Also, the basis of the numbers provided in the tree should be explained so that readers can easily search for the sequence data.

Reply: Several genomic sequences were used for each species to verify the absence of genetic intra-species variability. We observed two distinct sequences in *P. falciparum* (*P. falciparum* 1 and *P. falciparum* 2), explaining why we designed two primers pairs for *P. falciparum* amplification. We also verified the differentiation between the two *P. ovale* subspecies (*P. ovale curtisi* and *wallikeri*). Indeed, the origin of the genomic sequences used is missing.

Action: Below the title of Figure 1, several details have been added to explain our representation more clearly: “several genomic sequences (identified by their gene ID, <https://www.ncbi.nlm.nih.gov/gene>) for species belonging to the *Apicomplexa* phylum have been aligned in order to study the genetic diversity within each species. For *P. falciparum*, two genetic lineages were observed (*P. falciparum* 1 and 2). The separation of the two *P. ovale* subspecies (*P. ovale curtisi* and *P. ovale wallikeri*) was verified”.

16) Line 186. Change "parasitological forms" to "parasites".

Action: We changed it (lines 309 to 310, 764 and 766) for “blood-stage parasite forms”.

17) Line 208. The authors emphasize the two related species of *P. ovale* elsewhere in the MS, but lump these together here. If they wish to emphasize diagnosis of both species they need to distinguish the two throughout the MS. Alternatively, since very few *P. ovale* parasites were studied, assertions that the protocol can distinguish the two species should be modified.

Reply: Indeed, we have grouped *P. ovale* species together in the materials and methods section. Of the 10 *P. ovale*-positive patients in our cohort, as we only had subspecies identifications for 5 patients. We do not claim to be able to distinguish the 2 *P. ovale* subspecies, but our message is that we can amplify the 2 subspecies with our *P. ovale* qPCR (“comparing with the identification obtained by the French National Reference Center of malaria, the *P. ovale* qPCR assay was able to amplify *P. ovale wallikeri* (n = 1) and *P. ovale curtisi* (n = 4) infections”).

Action: In the “species confirmation” section of materials and methods section, we have added an explanation on *P. ovale* subspecies identification carried out by French National Reference Center of malaria: “in case of *P. ovale* target positivity, subspecies identification was carried out using in-house qPCR with high resolution melting” (lines 210 to 211).

18) Line 244. "The limit of detection....was detectable"? Correction is needed.

Reply: LOD determination of the qPCR *Plasmodium* spp. from a plasmid has been reported in the materials and methods “qPCR efficiency and limit of detection”.

Action: We simplified the sentence as follows: “the limit of detection of the *Plasmodium* spp. assay was 1 copie/μl (Supplemental Table 1)” (line 389).

19) Line 247. It is important that, for *P. ovale*, the two sub-species were not distinguished and, for *P. knowlesi*, only a culture isolate could be studied.

Reply: In the “qPCR efficiency and limit of detection” section of the materials and methods, we had specified that only *P. knowlesi* culture was used to verify the analytical performance of *P. knowlesi* qPCR. To be more precise, we have explained in Table 1 (sequences of the probes, forward and reverse primers designed on 18S rRNA) that we designed *P. ovale* primers and probe to amplify a common genomic sequence of *P. ovale curtisi* and *P. ovale wallikeri*.

Action: In Table 1, we added the following details (lines 986 to 988): “two reverse primers were developed to amplify the two distinct *P. falciparum* lineages (*P. falciparum* 1 and 2, Figure 1). One primer pair was designed on a common genomic region allowing amplification of both *P. ovale curtisi* and *P. ovale wallikeri* (with no distinction between them).”

20) Line 263. The text refers to "all 6 species", but in fact results for only 5 species are shown in Table 2 (with results for the two *P. ovale* sub-species lumped together).

Reply: For greater clarity and consistency, we will refer to 5 species (*P. falciparum*, *P. vivax*, *P. ovale*, *P. malariae* and *P. knowlesi*) in the manuscript.

Action: “six species” were replaced by “five species” throughout the manuscript.

21)a) Line 283 "Epidemiological or clinical data, available for 7 of them, were consistent with the detection of low plasmodial DNA load in blood (Table 3)." This sentence is unconvincing. Evidence for malaria in these patients is very limited. The authors should not claim here or in the abstract (lines 38-39) that there was convincing support for the diagnosis of malaria based on epidemiological information. Also, in Table 3, the meaning of "Two thick drops positive in Ivory Coast with malaria treatment." is unclear. Overall, it is possible that the patients with low positive qPCR readings had malaria infections, but also possible that these were false positives.

Reply: We agree that this part of the text lacked precision.

Action: The sentence "epidemiological or clinical data, available for 7 of them, were consistent with the detection of low plasmodial DNA load in blood (Table 3)" were deleted.

More details were added for BS-negative patients and qPCR-positive:

We have changed the sentence "two positive thick drops in Ivory Coast with antimalarial treatment" to "diagnosis and treatment of malaria based on two successive positive thick drops in Ivory Coast" (Table 3, line 992).

Positive qPCR results were checked a second time (lines 415 to 416). Confirmation of these results demonstrates the *Plasmodium* DNA detection in treated (probably in endemic areas) or in sub-microscopic malaria patients (lines 416 to 443):

- Patients 184 and 191 were diagnosed with malaria in malaria-endemic area with “positive thick drops”, and were therefore treated. Patient 322 had a “positive *P. falciparum* ICT” with a negative BS, probably corresponding to malaria treated in a malaria-endemic area.
- Patients 167 and 188 presented “fever” and digestive disorders on return from malaria-endemic area.

- As we had few epidemiological and clinical informations for patients 182 and 186, we assume that malaria research was motivated by the presence of fever.

21)b) Line 389. These episodes of recurrent *P. vivax/P. ovale* infection were presumably considered relapses, and this term, rather than "reactivation" should be used. Also, it would be helpful to know if these patients were treated with primaquine or tafenoquine for their primary infections.

Reply: Unfortunately, we do not have data on treatments by primaquine or tafenoquine after primary episode of infection. Indeed, these relapses are probably explained by the absence of treatment with these drugs.

Action: "reactivation" has been replaced by "relapse" throughout the manuscript. We added (lines 812 to 813) that primaquine or tafenoquine were not reported after the primary episode that denoted the absence of such treatment and then the later relapse.

22) Line 451. The precise meaning of "pauci-symptomatic" is unclear, and it is not known if any of the cases detailed in Table 3 represented smear negative symptomatic malaria. The Discussion should discuss potential benefits (improved sensitivity) and risks (decreased specificity) of a more sensitive assay to detect plasmodial DNA.

Reply: Indeed, we had two patients (patients 167 and 188, Table 3) for whom microscopic examination was negative with a positive qPCR on at least one of duplex qPCR targets. Both patients presented clinical symptoms consistent with recent malaria (fever, digestive disorders).

Action: Further details are provided in this paragraph. The term "pauci-symptomatic" has been removed, specifying that these are probably cases of submicroscopic malaria: "these results support the value of robust and sensitive molecular tools for the early diagnosis of sub-microscopic malaria cases" (lines 692 to 693).

Benefits were described in the discussion (lines 690 to 692) with better sensitivity of qPCR compared to microscopic examination, particularly for sub-microscopic malaria. qPCR sensitivity also represents a drawback, with detection of plasmodial DNA during post-treatment follow-up (lines 753 to 759) as with ICT.

23) Line 453. As noted above (point 14) it would be helpful here to explain the specific advantages of the new qPCR system described in this manuscript compared to others that have already been reported or are sold commercially.

Reply: The advantages of our duplex qPCR are:

- i) the use of pan-*Plasmodium* and *P. falciparum*-specific targets, combined with a search for qPCR inhibitors in the same reaction mixture,
- ii) the possibility of evaluating plasmodial copy number/ μL to estimate parasite load.

LAMP technique, which is the most widely used in french laboratories, does not allow species identification or parasite load assessment.

BIOSYNEX Ampliquick® Malaria qPCR is based on the same principle as qPCR developed in this manuscript. However, a French study carried out by the National Reference Center of malaria demonstrated a lower sensitivity of commercial BIOSYNEX Ampliquick® Malaria and Bio-Evolution *Plasmodium* Typage assays than LAMP-based assay Alethia® Malaria and in-house TaqMan qPCR assays in sub-microscopic malaria cases (Bouzayene A. *et al.*, Malar J., 2022).

It would be interesting to compare our qPCR with these two commercial assays, which target *18S rRNA* gene. Furthermore, it would also be interesting to compare our qPCR with that of French National Reference Center of malaria, which targets the *var* genes present in several copies in the *Plasmodium* genome. This work is planned and will be the object of a new publication in the future.

Action: The following sentence (lines 693 to 697) has been added to compare different assays available in France for malaria diagnosis by molecular biology: “A French study carried out by the National Reference Center of malaria showed a lower sensitivity of commercial assays (BIOSYNEX Ampliquick® Malaria and Bio-Evolution *Plasmodium* Typage assays) as compared to LAMP-based assay Alethia® Malaria and in-house TaqMan qPCR, particularly in negative microscopic malaria cases”.

24) Line 480. The parenthetical statement is not clear; it should be replaced by a sentence explaining the key point regarding vivax and ovale infections.

Reply: Indeed, we wanted to explain that it is important to diagnose mixed infections involving *P. vivax* and *P. ovale* in order to initiate appropriate treatment by primaquine or tafenoquine to prevent relapses.

Action: The following sentence (lines 741 to 745) has been added: “It is important to diagnose mixed infections involving *P. vivax* and *P. ovale* in order to initiate treatment by primaquine or tafenoquine for the prevention of relapse episodes.”

25) Lines 488 and 497. Replace "secondary", which implies causality, with "after."

Reply: Thank you for this comment.

Action: “secondary” has been replaced by “after” throughout the manuscript.

26) Line 496. The sentence beginning "The positivity of qPCR after anti-malarial treatment can be explained" is not clear and is incomplete. Likely the main explanations for persistent PCR identification after treatment are persistent parasite DNA for some days after parasite killing and circulating gametocytes, which are not killed by most therapies. Gametocytes are mentioned a few lines later, but the arguments should be consolidated into a simpler explanation.

Reply: Indeed, we had written two sentences that can be grouped together to explain qPCR positivity persistence.

Action: We have grouped (lines 762 to 765) together the two sentences to explain the persistence of positive qPCR: “qPCR positivity after anti-malarial treatment can be explained by better sensitivity of qPCR compared to conventional methods, elimination of parasite DNA after parasite death and sub-microscopic asexual blood-stage parasite or gametocytes persistence.”

27) Line 500. Change "parasitical" to "parasite".

Action: To maintain consistency we changed it for “blood-stage parasite forms” (lines 309 to 310, 764 and 766).

28) A limitations paragraph would be helpful. This could include the following limitations, and possibly others: a) few non-falciparum and no *P. knowlesi* clinical isolates were studied; b) clinical information for study subjects was very limited; c) the study was not equipped to identify false positive results from the qPCR assay (including clinical false positives for patients with circulating plasmodial DNA but no active infection).

Reply: Thank you for these suggestions. However, we would not use false-positive term to characterize positive qPCR in patients exposed to *Plasmodium* in endemic areas with negative microscopic examination and immunochromatographic test. Indeed, the qPCR results were verified a second time in these patients and we had the confirmation of PCR positivity, consistent with recent, sub-microscopic or treated malaria.

Action: We have added a section on study limits at the end of the discussion (lines 816 to 822): "Our study regarding these new *Plasmodium* qPCR assays have some limitations. Indeed, we had a limited sample size for non-*P. falciparum* species. Because of the limited number of imported *P. knowlesi* malaria cases in France, we were not able to test clinical samples to study properly the performance of the pan-*Plasmodium* and the specific *P. knowlesi* qPCR assays. Finally, we lacked clinical information to better describe and explain positive qPCRs in patients with negative results by conventional techniques. In addition, these samples were not checked with another molecular biology assay".

Response to Reviewer 2

The authors have developed a qPCR assay based on 18S rRNA gene and compared it with the outcomes of BS (considering it as a gold standard) & ICT. The LOD of the qPCR presented by the authors was 8 copies/reaction. It's not apparent what they mean by copies in this context. Are these copies of the target gene or the genome of the parasite or plasmid?

Further reactions may be of any volume, so the LOD would vary accordingly. Therefore, authors should be very precise when presenting such key outcomes. Moreover, as per authors control-plasmid was detectable until a dilution of 1 copies/ μ L & using clinical samples, the limit of detection was between 0.02 & 2.85 parasites/ μ L depending on the species. How this '8 copies/reaction' came into picture is not described or discussed anywhere in results.

In the abstract, the authors mentioned five cases of biparasitism (5.7%) by qPCR and concluded qPCR better than BS and ICT. However, in supplemental table-3 they presented 8 cases of biparasitism (9.2%) by BS. It is not clear how they concluded this outcome? Further in table 5, out of the 8 cases, only 2 were having biparasitism by BS. Such outcomes seem to be very confusing throughout the manuscript.

It is unclear whether each sample has been tested for all *Plasmodium* species through species-specific qPCR assay or not except 38 samples (positive for a single species by BS & ICT). Because that would be must to see the cross-reactivity.

Also, the qPCR outcomes were compared with BS which is very inferior (lower LOD) to qPCR. The actual results of clinical samples should be known before validating a method using those samples.

Although, the authors mentioned species confirmation was done by the real-time qPCR kit (French National Reference Center) but details (for example: based on which target gene & LOD) of that kit was not given.

Nucleic Acid Extraction: the authors mentioned they added an internal control per sample. It has not been detailed further why this addition was done? Authors simply have given a statement.

Validation of qPCR: Authors stated, 333 samples from 323 patients suspected of malaria but negative by BS and ICT were screened with duplex qPCR assay. Later, under "Analysis of negative samples on blood smears" they said one was positive by ICT. Such negligence creates doubt on the presented outcomes.

In table 3, sample negative for *Plasmodium* spp. & positive for *P. falciparum* is not justifiable. All species-specific positive samples must be positive for *Plasmodium* spp. Further, logically the Cq values of *Plasmodium* spp. should be less than that of species-specific Cq values but the authors observed just opposite.

1) It's not apparent what they mean by copies in this context. Are these copies of the target gene or the genome of the parasite or plasmid?

Reply: These are copies of copies of the target gene (*18S rRNA* gene). This copy number established using a calibration curve built using a plasmid containing the genomic sequence targeted by *Plasmodium* spp. qPCR (“quantitative qPCRs” in materials and methods).

Action: an explanation has been added to the “quantitative qPCRs” section in materials and methods (lines 302 to 303): “using a conserved *Plasmodium* locus across species of *18S rRNA* gene in a control-plasmid to implement calibration curve”.

2) How this '8 copies/reaction' came into picture is not described or discussed anywhere in results.

Reply: The copy number calculation has been fully described in the “quantitative qPCRs” section of the materials and methods.

Action: A 179 base pairs-plasmid containing a common locus of the *18S rRNA* gene of human *Plasmodium* species was produced at a concentration of 40 ng/μL, i.e. a concentration of 5.03×10^{10} copies/μL. Successive 10-fold dilutions were performed in order to establish a calibration curve for determination of copy number of the target locus/μL from Cq obtained by qPCR in the analyzed sample.

3) In the abstract, the authors mentioned five cases of biparasitism (5.7%) by qPCR and concluded qPCR better than BS and ICT. However, in supplemental table-3 they presented 8 cases of biparasitism (9.2%) by BS. It is not clear how they concluded this outcome? Further in table 5, out of the 8 cases, only 2 were having biparasitism by BS.

Reply: We thank reviewer 2 for this comment. The explanations were unclear and inconsistent. Cases of malaria involving two different species (n = 8) were detected by standard methods [BS or ICT] (n = 2/8), by the qPCR used by the French National Reference Center of malaria (n = 5/8), or with our qPCR assay (n = 5/8).

Action: Some clarifications have been made to the manuscript:

- Supplemental table 3: in the “positive BS” column, “mixed *Plasmodium* species infection” category has been clarified. “*diagnosis of mixed *Plasmodium* infection was performed by conventional methods (BS or ICT), by qPCR of French National Reference Center of malaria or by our qPCR assays”.
- Table 5: Title has been changed to “mixed *Plasmodium* species infection identified by conventional diagnostic methods and/or molecular biology”.
- In the abstract, sentence on mixed infections has been modified: “of the 8 cases of mixed *Plasmodium* species infections, 5 were identified with our qPCR assays with better sensitivity as compared to BS and ICT”.

4) It is unclear whether each sample has been tested for all *Plasmodium* species through species-specific qPCR assay or not except 38 samples (positive for a single species by BS & ICT).

Reply: The sentence “these positive samples were analyzed with all qPCRs developed” lacks precision. Samples from *P. falciparum*-positive patients were analyzed using duplex qPCR

(*Plasmodium* spp. and *P. falciparum*). Patient samples positive for *P. vivax*, *P. ovale* and *P. malariae* were analyzed using duplex qPCR and the species-specific qPCRs

Action: the sentence “these positive samples were analyzed with all qPCR developed” was replaced by “samples from *P. falciparum*-positive patients were analyzed using the duplex qPCR assay. Samples positive for *P. vivax*, *P. ovale* and *P. malariae* were analyzed using duplex qPCR and species-specific qPCR assays” (lines 341 to 343).

5) Also, the qPCR outcomes were compared with BS which is very inferior (lower LOD) to qPCR. The actual results of clinical samples should be known before validating a method using those samples.

Reply: Results of positive samples were confirmed by the National Reference Center of malaria, both by microscopic techniques and by molecular biology (“species confirmation” in materials and methods). Practically speaking, the results were available before the evaluation of the qPCR assays began.

However, a limitation of the study could be that negative samples by blood smears and positive with our qPCR were not verified with another molecular biology assay.

Action: This point has been added to the discussion (lines 821 to 822): “In addition, these samples were not checked with another molecular biology assay”.

6) Although, the authors mentioned species confirmation was done by the real-time qPCR kit (French National Reference Center) but details (for example: based on which target gene & LOD) of that kit was not given.

Reply: We thank the reviewer for this comment. These data were missing from the materials and methods.

Action: The genomic target and sensitivity have been added to paragraph (lines 207 to 210): “the *Plasmodium* Typage (Bio-Evolution, Bussy-Saint-Martin, France) real-time qPCR kit has been used for simultaneous identification of *P. falciparum*, *P. ovale*, *P. vivax*, *P. malariae* and *P. knowlesi* targeting *18S rRNA* gene with sensitivity of 10 copies/ μ L”.

7) Nucleic Acid Extraction: the authors mentioned they added an internal control per sample. It has not been detailed further why this addition was done? Authors simply have given a statement.

Reply: The internal control was added at the extraction step to control the extraction step, the absence of qPCR inhibitors and the exactitude of the pipetting. This is a standard procedure recommended by the MIQE guidelines (Bustin SA, Clin Chem., 2009).

Action: The following sentence (lines 229 to 230) has been added: “An internal control added before the extraction step was used to control all the qPCR process as recommended by the MIQE guidelines”.

8) Validation of qPCR: Authors stated, 333 samples from 323 patients suspected of malaria but negative by BS and ICT were screened with duplex qPCR assay. Later, under "Analysis of negative samples on blood smears" they said one was positive by ICT. Such negligence creates doubt on the presented outcomes.

Reply: We warmly thank the reviewer because this was a mistake. All 333 samples were negative by BS. 332 samples were negative by BS and ICT, and one sample was negative by BS and positive by ICT.

Action: The sentence has been modified (lines 344 to 345): "in addition, 333 samples negative by BS from 323 patients suspected of malaria were screened with duplex qPCR assay".

9) In table 3, sample negative for *Plasmodium* spp. & positive for *P. falciparum* is not justifiable. All species-specific positive samples must be positive for *Plasmodium* spp. Further, logically the Cq values of *Plasmodium* spp. should be less than that of species-specific Cq values but the authors observed just opposite.

Reply: 2 qPCR assays based on different primers and different DNA targets have different characteristics including different efficiencies (here the *Plasmodium* spp. assay have a decreased efficiency as the specific of *P. falciparum*) but also different behaviors regarding the DNA amplification depending on the secondary structures of the extracted DNA, the GC content or the target for example. All the chemicals and physical properties of each PCR amplification cannot be standardized and cannot be anticipated. This is exactly why different our assays have different performances. In addition, when the target quantity is low, qPCR is even more versatile as it responds to the Poisson's law, meaning that pipetting will not take every time the same amount of copies of target DNA by chance explaining why sometimes only one out of 3 replicates will be positive when the limit of detection is close. This can explain perfectly why an assay can be positive an another negative for a given sample.

Counter intuitively, pan assays are frequently less sensitive and less specific than specific assays that have frequently higher sensitivity. This had been already explained in specific papers including ours (Alanio A. and Bretagne S., Clin Microbiol Infect., 2014).

Action: We added to the discussion few sentences to explain these discrepancies (lines 685 to 690) as: "Amplifications with *P. falciparum*-specific qPCR were earlier than by *Plasmodium* spp. qPCR. Genus-targeted tests are often less sensitive than species-specific assays (Alanio A. and Bretagne S., Clin Microbiol Infect., 2014). One patient had a positive *P. falciparum*-specific qPCR and a negative *Plasmodium* spp. qPCR (patient 188, Table 3). This discrepancy may be due to a low quantity of genomic target with random amplification responding to Poisson's law, or a better efficiency of specific-*P. falciparum* qPCR".

Re: Spectrum01622-24R1 (**Implementation and validation of a new qPCR assay to detect imported human *Plasmodium* species.**)

Dear Dr. Camille Cordier:

Thank you for the privilege of reviewing your work. Below you will find my comments, instructions from the Spectrum editorial office, and the reviewer comments.

As noted by Reviewer one, false-positives cannot be excluded based on patient symptoms consistent with malaria, travel to a malaria endemic area, and repeat qPCR results for the new assay. A stronger statement needs added to the discussion stating that false-positive results cannot be excluded.

Revision Guidelines

Sincerely,
Wendy Szymczak
Editor
Microbiology Spectrum

Dear Wendy Szymczak,

We thank you for the reviewing of our manuscript “Implementation and validation of a new qPCR assay to detect imported human *Plasmodium* species” to Spectrum.

We added a sentence in discussion regarding possible false-positive qPCRs results.

Please find attached the Marked-Up Manuscript and the response to reviewers.

We thank you for considering our revised manuscript,
Sincerely,
Dr. Camille Cordier and Pr. Alexandre Alanio.

Response to Reviewer

As noted by Reviewer one, false-positives cannot be excluded based on patient symptoms consistent with malaria, travel to a malaria endemic area, and repeat qPCR results for the new assay. A stronger statement needs added to the discussion stating that false-positive results cannot be excluded.

Reply: In the study conducted by Taylor S. *et al.* (doi: 10.1128/JCM.00565-14) of 10 samples from 9 centers, two results were false-positive for *P. falciparum* target. Both results were detected with duplex and multiplex qPCRs targeting 18S rRNA.

Action: The following sentence has been added to the discussion (line 338 to 340): “Despite presence of symptoms suggestive malaria, travel to malaria-endemic areas and qPCR duplex results verifications, false-positive results cannot be formally excluded for these patients”.

Re: Spectrum01622-24R2 (**Implementation and validation of a new qPCR assay to detect imported human *Plasmodium* species.**)

Dear Dr. Camille Cordier:

Your manuscript has been accepted, and I am forwarding it to the ASM production staff for publication. Your paper will first be checked to make sure all elements meet the technical requirements. ASM staff will contact you if anything needs to be revised before copyediting and production can begin. Otherwise, you will be notified when your proofs are ready to be viewed.

Sincerely,
Wendy Szymczak
Editor
Microbiology Spectrum